# Adeno-associated virus delivered CXCL9 sensitizes glioblastoma to anti-PD-1 immune checkpoint blockade

Christina A. von Roemeling [1,2] ✉, Jeet A. Patel[1,2], Savannah L. Carpenter[1,2], Oleg Yegorov[1,2], Changlin Yang[1,2], Alisha Bhatia[1,2], Bently P. Doonan [2,3], Rylynn Russell[1,2], Vrunda S. Trivedi[1,2], Kelena Klippel[1,2], Daniel H. Ryu[4], Adam Grippin [5], Hunter S. Futch[6], Yong Ran[4], Lan B. Hoang-Minh[1,2], Frances L. Weidert[1,2], Todd E. Golde[4] & Duane A. Mitchell [1,2] ✉

There are numerous mechanisms by which glioblastoma cells evade immunological detection, underscoring the need for strategic combinatorial treatments to achieve appreciable therapeutic effects. However, developing combination therapies is difficult due to dose-limiting toxicities, blood-brain-barrier, and suppressive tumor microenvironment. Glioblastoma is notoriously devoid of lymphocytes driven in part by a paucity of lymphocyte trafficking factors necessary to prompt their recruitment and activation. Herein, we develop a recombinant adeno-associated virus (AAV) gene therapy that enables focal and stable reconstitution of the tumor microenvironment with C-X-C motif ligand 9 (CXCL9), a powerful call-and-receive chemokine for lymphocytes. By manipulating local chemokine directional guidance, AAV-CXCL9 increases tumor infiltration by cytotoxic lymphocytes, sensitizing glioblastoma to anti-PD-1 immune checkpoint blockade in female preclinical tumor models. These effects are accompanied by immunologic signatures evocative of an inflamed tumor microenvironment. These findings support AAV gene therapy as an adjuvant for reconditioning glioblastoma immunogenicity given its safety profile, tropism, modularity, and off-the-shelf capability.

Breakthroughs in immunotherapy including immune checkpoint blockade (ICB), monoclonal and bispecific antibodies, and CAR T cell therapy, have ignited the hope of achieving durable remission in even the most recalcitrant tumors. Although these strategies are capable of producing remarkable responses, therapeutic benefit is seen in only a small proportion of patients with many proposed reasons for lack of response[1]. Presently, T cell infiltration and abundance within the tumor microenvironment is one of the most predictive biomarkers for response to immunotherapy[2-4]. Unfortunately, diseases like glioblastoma (GBM) and many other solid tumors demonstrate low baseline infiltration of lymphocytes that is only marginally improved by treatment[5,6]. For GBM patients in particular, generalized lymphopenia as a result of impaired lymphocyte egress[7] alongside the lymphodepleting nature of conventional treatment further detracts from

[1]Lillian S. Wells Department of Neurosurgery, University of Florida, Gainesville, FL, USA. [2]Preston A. Wells, Jr. Center for Brain Tumor Therapy, University of Florida, Gainesville, FL, USA. [3]Department of Medicine, Hematology and Oncology, University of Florida, Gainesville, FL, USA. [4]Goizueta Brain Health Institute, Emory University School of Medicine, Atlanta, GA, USA. [5]Department of Radiation Oncology, MD Anderson Cancer Center, The University of Texas, Houston, TX, USA. [6]Department of Neurosurgery, Emory University School of Medicine, Atlanta, GA, USA. ✉e-mail: christina.vonroemeling@neurosurgery.ufl.edu; duane.mitchell@neurosurgery.ufl.edu

successful adaptive immune recognition of these tumors[8]. Preclinical studies on cellular therapy imaging and trafficking show that a comparably lower fraction of T cells can be found in these brain tumors as compared to lung, liver, and spleen tissues in comparative mouse models[9,10]. Similar hurdles with respect to CAR T cell penetration into brain tumors in the clinical setting have necessitated alternative locoregional delivery strategies[11]. Together these observations indicate that ineffective T cell migration and infiltration into GBM tumors may represent a principal barrier to immunotherapy. Solving this problem requires both an improved understanding of the chemical signals that govern T cell chemotaxis into tumors and identifying a method to amplify those signals.

Lymphocytes are recruited via long-range signaling mediated by the diffusion of chemokines present in inflammatory environments. During glioma formation, tumors manufacture immune-suppressive chemokines and cytokines that co-opt resident cells, resulting in the preferential recruitment of immune suppressor cells from the periphery[12]. Additionally, we have found that primary patient glioma samples are deficient in lymphocyte-specific chemokines. To overcome this problem, we propose the use of in situ recombinant AAV expressing a lymphocyte trafficking chemokine payload to restore expression of lymphocyte chemotaxis. AAV vectors are the leading platform for gene delivery due to their efficacy, ease of use, and safety profile[13]. Additionally, multiple AAV gene therapies have reached FDA approval for a variety of diseases[13,14]. There are multiple strategies for AAV therapy for GBM that have been previously described, including vector capsid protein modification, brain tropic (microglia, astrocyte, or neuronal) AAV serotypes, and direct parenchymal delivery of AAV[15–19]. Thus, AAV vectors are an ideal platform for transforming targeted cell populations while minimizing potential patient risk.

In this work we identify CXCL9 as a candidate lymphocyte call-and-receive signal absent in GBM following a comprehensive chemokine screen of clinical tumor specimens. We examine AAV delivered CXCL9 transgene tropism, durability, and impact on lymphocyte tumor migration. Using syngeneic preclinical model systems of GBM, we further evaluate the therapeutic benefit of our AAV gene therapy alongside anti-PD-1 ICB. Single-cell RNA sequencing (scRNAseq) of immune cells isolated from tumors during treatment reveals widespread immunological reconditioning of tumors, with improved effector lymphocyte recruitment that yields long-term survival outcomes in aggressive preclinical GBM models. This work suggests direct conditioning of the tumor microenvironment by AAV-CXCL9 constructs could potentially overcome a key resistance factor in GBM, sensitizing tumors to immune-mediated therapies.

## Results

### Chemokine analysis of Human GBM favors MDSC recruitment over lymphocyte infiltration

Upon examination of human glioma tumors for CD3 protein expression provided through the Human Protein Atlas, we found that > 80% of tumor specimens evaluated are negative for lymphocyte infiltrates (Supplementary Fig. S1a), corroborating prior literature describing the lymphocyte-replete nature of these tumors[20]. To identify if a paucity of lymphocyte chemotactic factors is a contributing aspect, human glioma samples were screened via chemokine proteome array. Of these, CXCL4, CXCL7, CXCL8 (IL-8), CXCL16, LCF (IL-16), TIG-2, and midkine (MDK) emerged as the most abundant secreted chemokines detected (Fig. 1a, b), where these ligands play a significant role in recruiting myeloid-derived suppressor cells in the context of gliomagenesis[21]. Notably, chemotactic factors that favor lymphocyte recruitment were poorly expressed, including CXCL9 (MIG) and MIP-1α/β (CCL3/CCL4) (Fig. 1a, b). CXCL9 is a powerful attractant known to induce the migration, differentiation, and activation of cytotoxic T lymphocytes[22]. Expression of CXCL9 has been shown to correlate with anti-tumor immune activity and is predictive of response to ICB in

several solid tumors[23]. We hypothesized that AAV delivery and restoration of CXCL9 as a "call-and-receive" signal for T lymphocytes within the tumor microenvironment (TME) would enhance their recruitment and infiltration in the tumor.

Historically, targeted transduction of cancer with AAV has proven challenging despite elegant efforts in capsid evolution studies and capsid engineering. To achieve sufficient transgene expression, transduction of either tumor cells or tumor-associated stroma is likely necessary, where intra-tumoral delivery would minimize the potential for systemic toxicities and decrease off target homing of T cells. To identify an appropriate AAV capsid for targeting glioma tumors, we performed an in vitro capsid screen. Enhanced green fluorescent protein (EGFP) was encoded into an AAV2 single stranded vector, utilizing the non-cell-autonomous, constitutively active CBA promoter to drive transgene expression (AAVn-EGFP). These constructs were pseudotyped into 29 unique capsids as previously described[24]. Transduction in 15 unique glioma models, including primary human[25] and murine xenografts, was assessed via EGFP relative fluorescence expression (Supplementary Fig. S1b). AAV6 was selected for further examination as it demonstrates moderate to high transduction in nearly all models tested and is further substantiated by excellent CNS transduction in other studies[24,26]. AAV6 capsids encoding each CXCL9, EGFP, blue fluorescent protein (BFP), and empty vector control (Fig. 1c, Supplementary Fig. S1c–e) were designed for further testing. Quantitative flow cytometric evaluation of EGFP expression in three distinct syngeneic murine GBM models 72 h following AAV6-EGFP transduction shows moderate transduction of GL261 and KR158, with > 25% of cells positive for the transgene at this time point, with lower transduction observed in CT-2A cells (<20%) (Supplementary Fig. S1f).

### AAV6 transduces tumor-reactive astrocytes in vivo in preclinical models of GBM

While we demonstrate good targeting of glioma cells in an in vitro setting, AAV6 has also been reported capable of targeting other cell populations in the CNS[24,26]. These studies have largely been performed in the context of naïve mice or in models of neurodegenerative disease, which may or may not be directly applicable to CNS malignancy. To define AAV6 tropism in murine GBM in vivo, AAV6-EGFP (Supplementary Fig. S1c) was intratumorally injected into established intracranial KR158 and GL261 tumors. EGFP expression was detected 1 week following tumor transduction in both models (Supplementary Fig. S2a, b), however the morphological appearance and contiguous distribution of transduced cells suggest that AAV6 targeted cells are likely tumor-associated, and not cancer cells directly. Microglia and tumor-associated macrophages are reported to comprise a significant cellular proportion of glioma tumors[27], and so we sought to identify if AAV6 was targeting either population. KR158 and GL261 were implanted into CCR2[RFP]CX3CR1[GFP] (B6.129(Cg)-*Cx3cr1*[tm1Litt] *Ccr2*[tm2.1Ifc]/JernJ) dual reporter mice, where microglia can be identified via GFP expression, and bone marrow-derived inflammatory cells via RFP expression[28,29]. Tumors were evaluated by 3D IHC for viral transduction 1 week following intratumor injection with AAV6 encoding a BFP reporter (Supplementary Fig. S1d). In both tumor model systems, we found no co-localization between BFP and either RFP or GFP, indicating that neither microglia nor tumor-associated macrophages are the principal target of AAV6 transduction (Supplementary Fig. S3a, b). To assess the degree of AAV6 transduction specifically in tumor cells we implanted mice with RFP-labeled KR158 or GL261 cells. Tumors were evaluated by 3D IHC for viral transduction 1 week following intratumor injection with AAV6-EGFP. Resected tumors were immuno-labeled against glial fibrillary acidic protein (GFAP) to detect astrocytes, another candidate tumor-associated cell population. Both tumor models reveal a high degree of overlap between GFAP (red) and EGFP (green), with minimal overlap between tumor cells (gray) and EGFP (Fig. 1d, Supplementary Fig. S3c), indicating that EGFP-positive cells are likely astrocytes. Voxel-

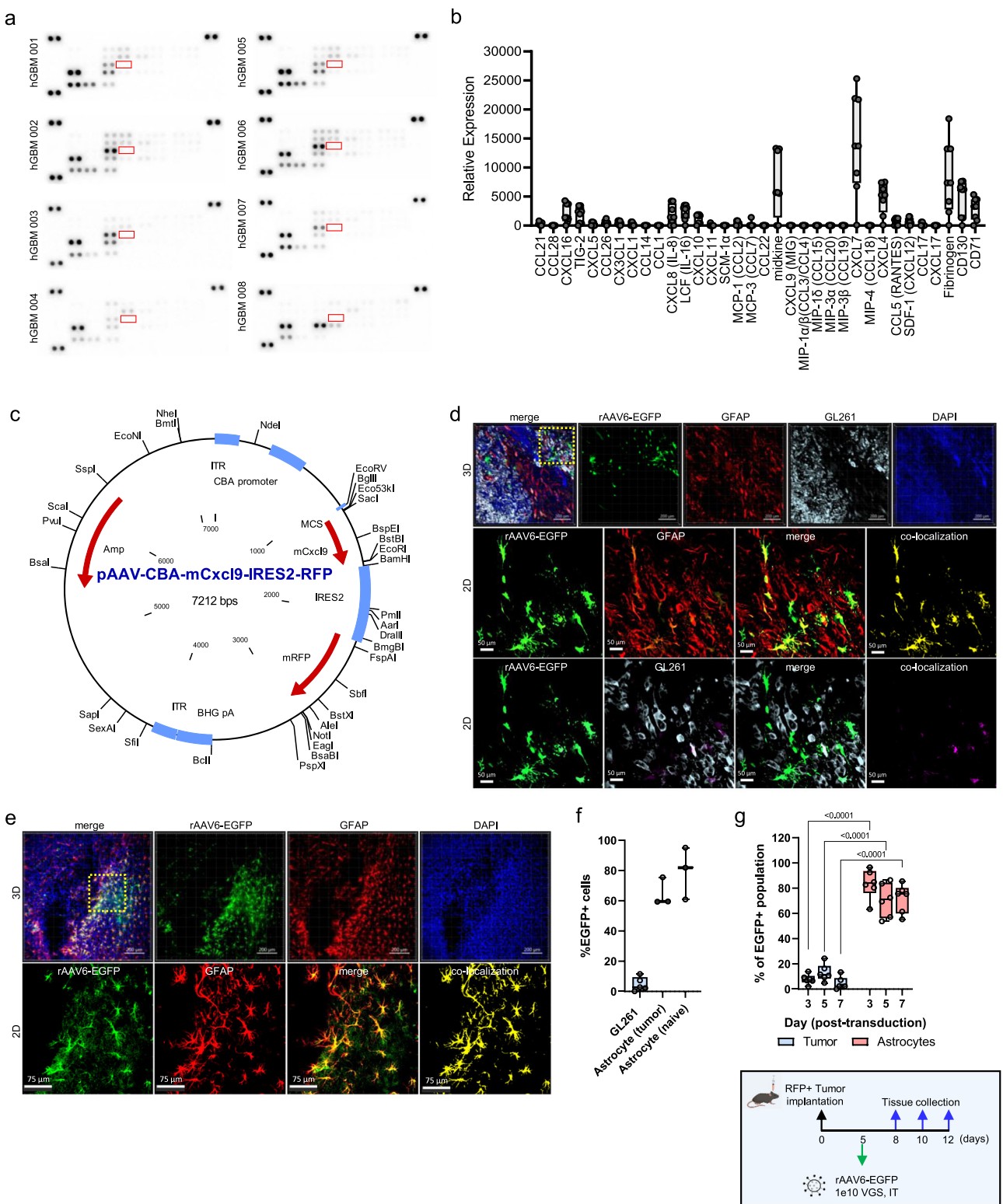

based co-localization algorithms to quantitate EGFP co-localization with each tumor or astrocytes confirm astrocytes as the principal cell target of AAV6 transduction, accounting for ~ 60–70% of EGFP-positive cells in both GL261 (Fig. 1f) and KR158 (Supplementary Fig. S3d) intracranial tumors. Because tumor presence can stimulate different activation states in astrocytes that may cause them to be more or less susceptible to viral transduction[30,31], we also evaluated CNS tropism of AAV6 in age-matched naïve mice. AAV6 was equally efficient at transducing astrocytes in naïve animals as shown by co-localization

between GFAP immunostaining and EGFP transgene expression (Fig. 1e, f).

One of the unique features of AAV gene transduction is that it rarely integrates into the host genome. Following uncoating in the host nucleus, single-stranded genomes are converted to double-stranded multimeric circular concatemeric episomes[32]. As such, AAV transgene expression can persist long-term in post-mitotic cells. Because tumor cells undergo rapid cell division, it may be possible that transgene expression is lost over time through sequential dilution of episomes

**Fig. 1 | Chemokine signature of glioblastoma tumors. a** Immunoblots of GBM samples showing signal intensity of 31 chemokines ($n = 8$). CXCL9 (undetected) is outlined in red. **b** Box-whisker plots of cumulative relative protein expression of immunoblots shown in panel (**a**). **c** Recombinant AAV6 vector design encoding CXCL9 and the fluorescent reporter gene RFP. **d** 3D IHC of RFP-labeled GL261 tumor tissue collected 1 week following AAV6-EGFP injection. The top row depicts 3D rendering at 10 x magnification, scale bar 200 μm. AAV6 transduced cells are shown in green, GFAP in red, RFP+ tumor cells in gray, and DAPI nuclear stain in dark blue. 2nd and 3rd rows depict 2D digital zoom as outlined by the yellow dashed line in the top row to enhance cellular resolution. Voxel-based co-localization between AAV6 and GFAP (2nd row) and AAV6 and tumor cells (3rd row) is shown as a separate channel (yellow or pink). Representative images selected from $n = 5$. **e** 3D IHC of AAV6-EGFP transduction in age-matched naïve control mice. The top row depicts 3D rendering at 10x magnification, scale bar 200 μm. AAV6 transduced cells are shown in green, GFAP in red, and DAPI nuclear stain in dark blue.

2nd row depicts 2D digital zoom as outlined by the yellow dashed line in the top row to enhance cellular resolution. Voxel-based co-localization between AAV6 and GFAP is shown as a separate channel (yellow). Representative images selected from $n = 3$. **f** Box-whisker quantitative summary of voxel-based AAV6 co-localization with either tumor (GL261, $n = 5$) or astrocytes in each tumor-bearing ($n = 3$) and naïve mice ($n = 3$). **g** Box-whisker plot of flow cytometry quantitation of AAV6 (EGFP +) co-localization with either tumor cells (GL261, RFP +) or astrocytes (GFAP + , RFP-) at 3-, 5-, and 7 days post AAV6 transduction as illustrated in the schematic. Two-way ordinary ANOVA statistical analysis performed comparing percent transduction between tumor and astrocytes across matched time points, $n = 6$ per time point. $P$-values ≤ 0.05 are considered statistically significant. Box-whisker plots display the box ranging from the first to the third quartile, the center median value, and the whiskers extend from each quartile to the minimum and maximum values. Source data are provided as a Source Data File.

passed down to daughter cells. To explore this, we evaluated AAV6-EGFP transgene expression longitudinally across early time points in mice harboring RFP labeled GL261 cells, which demonstrate the highest transduction efficiency in vitro (Supplementary Fig. S1f). Tumors were resected 3, 5, and 7 days following AAV6-EGFP intra-tumor injection as outlined in Fig. 1g, and EGFP transgene expression in tumors and astrocytes was measured by flow cytometry. Even at early time points, AAV6 predominantly transduces astrocytes identified as GFAP + RFP- (70–80% EGFP+ cells), with limited EGFP expression observed in RFP+ tumor cells (<15%) (Fig. 1g, Supplementary Fig. S4a). By 7 days post intratumor viral injection, less than 5% of EGFP+ cells on average were RFP+ tumor cells, where astrocytes consistently comprised 70–80% of EGFP+ cells at each time point. These data indicate that AAV6 more selectively transduces astrocytes in vivo, with limited and transient expression in tumor cells.

## AAV6 transgene signal distribution and durability in GBM
Next, we examined the distribution of transgene signal in both the GL261 and KR158 tumor models via 3D IHC to better understand the avidity of AAV6 for tumor-associated versus distal astrocytes following direct intra-tumor injection. BFP transgene expression was observed in a peritumoral pattern in and around the tumor body in both model systems (Fig. 2a, b), redolent of glial scar formation found in human brain malignancies[33]. As CXCL9 is a small, secreted chemokine, we wanted to determine if signal expression was still focal to the tumor or could be detected in contralateral brain and/or systemically. The whole brain was collected at 1 and 2 weeks following AAV6-CXCL9 or AAV6-EGFP intratumor injection as outlined in Fig. 2c. Cerebellar tissue was removed, and remaining tissue was dissected into the tumor containing and contralateral hemispheres. Serum was collected following peripheral blood draws taken from the posterior vena cava. Brain tissue and serum were also collected from non-transduced (sham) tumor controls, and naïve (non-tumor bearing) controls to establish CXCL9 baseline values. Serum levels of CXCL9 following intratumor delivery of AAV6-CXCL9 measured using high sensitivity ELISA assay did not exceed those observed in naïve controls (Fig. 2d, e). In the brain, elevated CXCL9 expression was selectively detected in the tumor bearing hemisphere transduced with AAV-CXCL9, with minimal signal observed in the contralateral hemisphere in both GL261 and KR158 model systems (Fig. 2f, g). Transgene CXCL9 expression appears to be stable, as signal intensity was consistent in AAV6-CXCL9 transduced tumors at both the 1- and 2-week time points in each tumor model (Fig. 2f, g). Of note, a small increase in CXCL9 expression was observed in AAV6-EGFP control transduced GL261 tumors and could be indicative of a mild inflammatory response to AAV6, however these values were not found to be statistically significant. Together these data demonstrate that AAV6 intratumor delivery of CXCL9 results in focal and durable expression of encoded transgene, where tumor-reactive astrocytes are the target of AAV6 transduction.

## AAV6-CXCL9 enhances lymphocyte chemotaxis
To evaluate the biologic activity of AAV6-CXCL9 on lymphocyte recruitment, we performed competitive in vitro chemotaxis assays. Briefly, CTV-labeled splenic-derived T lymphocytes were flanked by target cells transduced with AAV6 encoding either EGFP or CXCL9, and migration was monitored via fluorescence microscopy at 1- and 24 h following co-culture as described in the methods (Fig. 3a). Using GL261 tumor cells as the target population for AAV6 transduction, significantly more T lymphocytes co-localized in the CXCL9 transduced tumor field as compared to EGFP at 24 h (Fig. 3b). Given that astrocytes are the principal target of AAV6 transduction in vivo, chemotaxis was reassessed via competitive co-culture using astrocytes (C8-D1A) in lieu of GL261 glioma cells. Astrocytes transduced with AAV6-CXCL9 similarly showed enhanced recruitment of T lymphocytes (Fig. 3c), confirming that transgene encoding CXCL9 produces a biologically functional chemokine. We also compared lymphocyte chemotaxis in the context of AAV6-CXCL9 transduced tumor cells versus AAV6-CXCL9 transduced astrocytes, and found that lymphocytes were evenly distributed in both the astrocyte field and tumor field (Supplementary Fig. S5a), even though increased levels of CXCL9 secretion were detected in the tumor field (Supplementary Fig. S5b). This we attribute in part towards the higher proliferative capacity of tumor cells as compared to C8-D1A astrocytes (Supplementary Fig. S5c). Altogether, these data indicate that CXCL9 functions in an autonomous manner, effectively promoting lymphocyte chemotaxis independent of the cellular source. It also suggests that there is a minimum biological threshold of CXCL9 signal needed for lymphocyte chemotaxis, beyond which increasing levels of this chemokine may not impart a competitive advantage.

To determine the effect of AAV6-CXCL9 on T lymphocyte recruitment in vivo, multiparametric flow cytometry was performed to quantitate the number of T cells present in dissociated tumors following intratumor delivery. These studies were done in combination with anti-PD-1 ICB, where tissue was collected 1 day following the final dose of ICB to capture events within the therapeutic response window as outlined in Fig. 3d. In both GL261 and KR158 tumor models AAV6-CXCL9 alone had minimal impact on enhancing T cell recruitment to the tumor, however significant increases in T lymphocyte infiltration were observed in the context of combination treatment. AAV6-CXCL9 plus ICB increased CD8 T lymphocytes > 2.5-fold in the GL261 model and >4.5-fold in the KR158 model (Fig. 3e, f and Supplementary Fig. S4b). While no significant changes in CD4 T lymphocyte recruitment in response to treatment was observed in the GL261 model (Fig. 3g), a >3-fold increase was detected in the KR158 model (Fig. 3h). Anti-PD-1 ICB treatment in combination with control AAV6 (EGFP) modestly increased CD8 T lymphocyte recruitment in GL261 by 1.4-fold and in KR158 by 2.7-fold, indicating that CXCL9 markedly improves tumor infiltration by these cells. These data highlight a potential role for anti-PD-1 ICB in mobilizing T lymphocytes systemically, where sequestration of T lymphocytes was recently proposed as

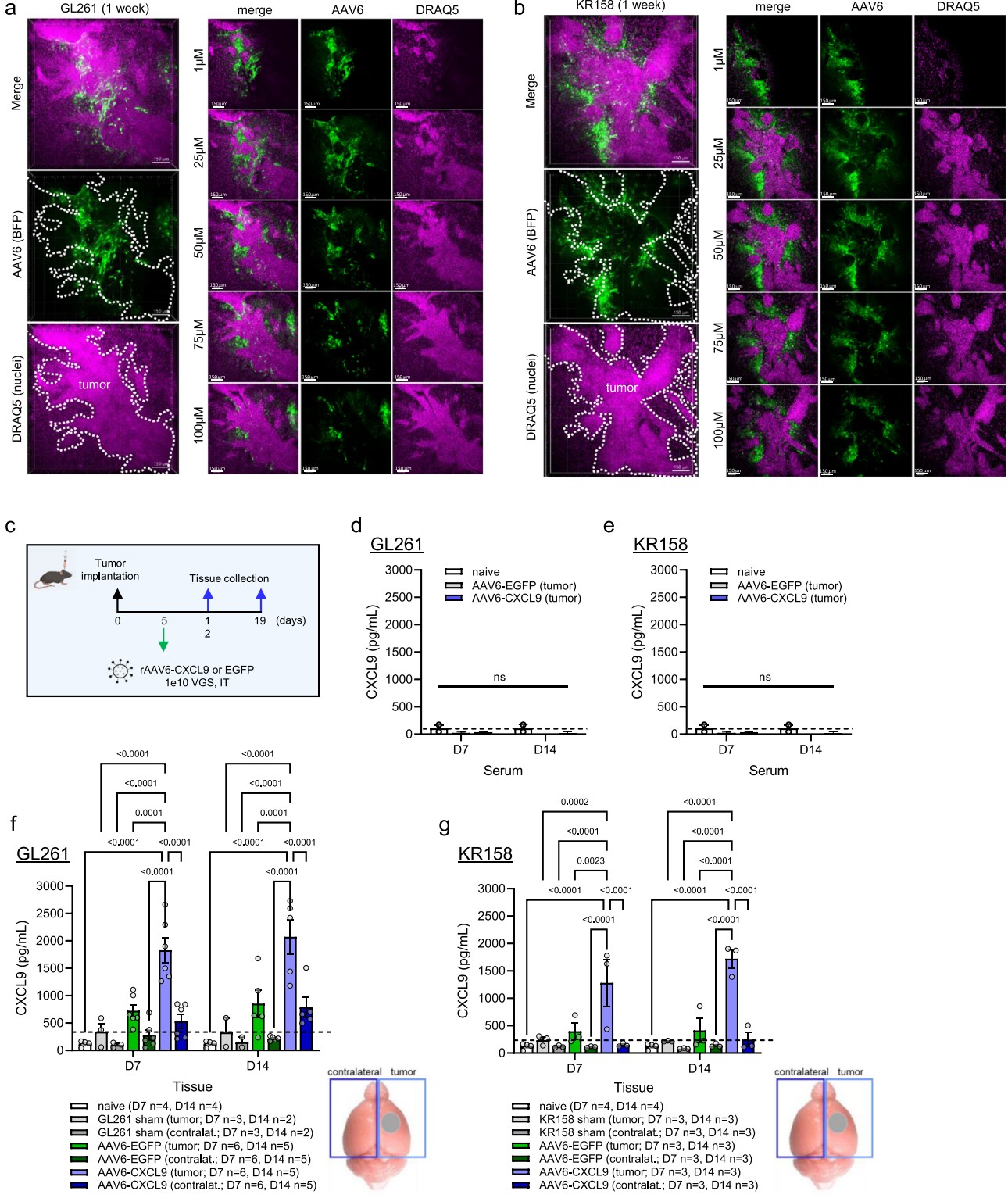

**Fig. 2 | AAV6 tumor tropism.** 3D IHC displaying geospatial distribution of AAV6 encoded transgene (BFP) in (**a**) GL621 and (**b**) KR158 tumors collected 1 week following in vivo transduction (green), $n = 2$ per model, scale bar 150 μm. DRAQ5 nuclear dye (pink) is used to identify tumor borders, as outlined by the white dashed line. **c** Intra-tumor AAV6 treatment schematic for protein detection of AAV6 encoded CXCL9. ELISA detection of CXCL9 protein in serum at one and two weeks following AAV6-CXCL9 or AAV6-EGFP control intracranial injection in (**d**) GL261 and (**e**) KR158 models, $n = 3$ per time point, per group. Age-matched naïve controls used to establish baseline CXCL9 levels indicated by dashed black line. ELISA detection of CXCL9 protein in brain tissues isolated at 1 and 2 weeks following AAV6-CXCL9 or AAV6-EGFP control intracranial injection in (**f**) GL261 and (**g**) KR158 models. Left and right hemispheres lysed separately to reflect tumor-bearing and contralateral (focal and distal) signal detection. Statistical analyses performed using two-way ANOVA analysis with Tukey's multiple comparisons test. Age-matched naïve brain, and sham (saline) injected tumors included as negative control and tumor baseline control, with the latter represented by the dashed black line, $n$ values shown. $P$-values ≤ 0.05 are considered statistically significant. Bar graphs depict group mean with error bars representing standard error of the mean. Source data are provided as a Source Data File.

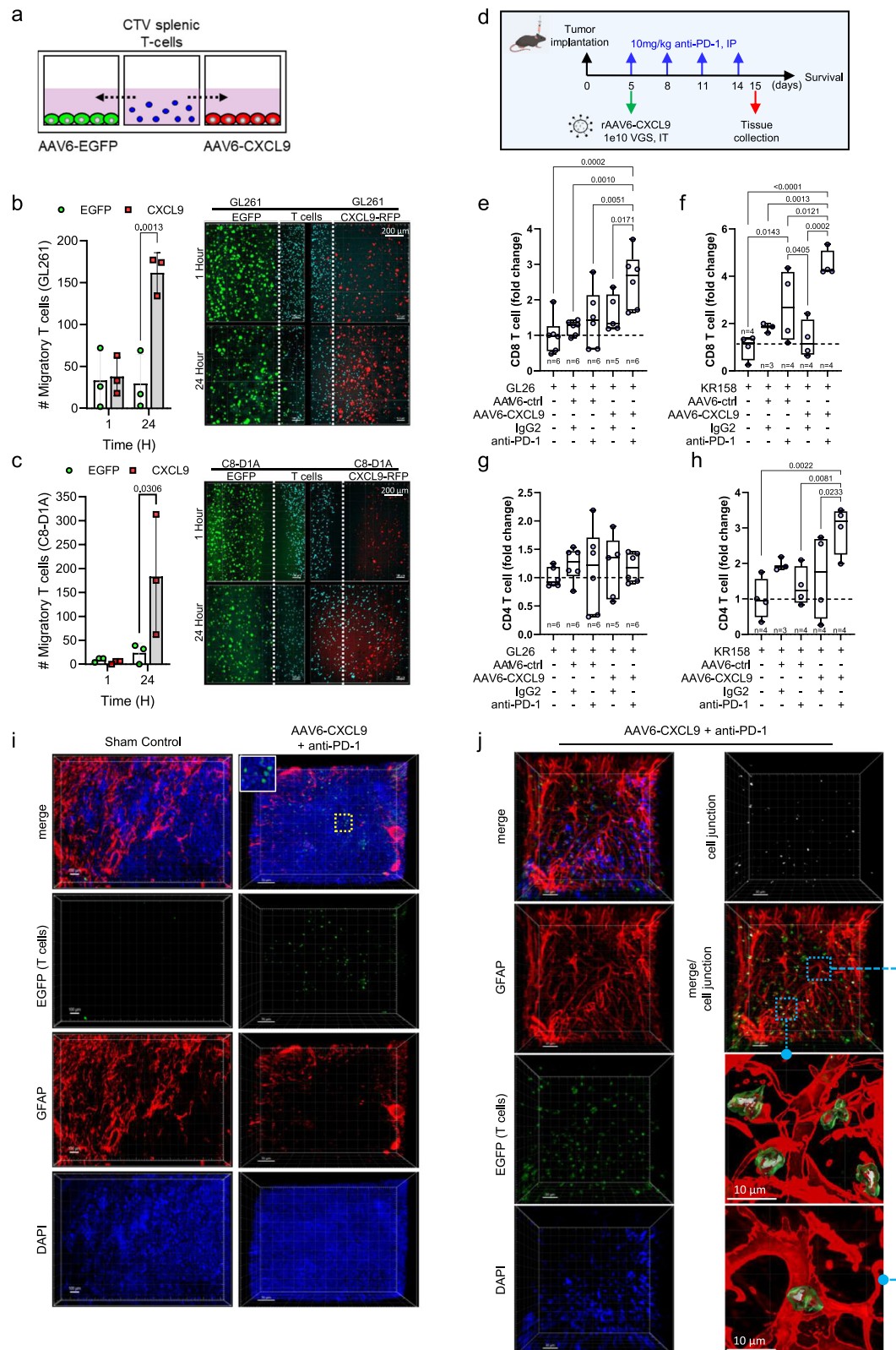

a mechanism of immune suppression in brain tumors[7]. To evaluate the geospatial distribution of T lymphocyte infiltration into treated tumors, we resected tumor tissue on day 15 following treatment with either sham control or combination AAV6-CXCL9 with anti-PD-1 ICB, where lymphocytes are identifiable through endogenous EGFP expression. Few, if any, lymphocytes could be visualized in control treated tumors (Fig. 3i). A heterogenous distribution of lymphocytes

could be readily detected in both peritumor and intratumor regions in combination treated tissue (Fig. 3i), confirming that treatment increases both lymphocyte tumor recruitment and penetration. In combination treated tumors, T lymphocytes were also observed in regions of astrocytosis, where they formed direct cell-cell contact points, or 'cell junctions (white),' as detected by voxel co-localization between GFAP (astrocytes, red) and EGFP (lymphocytes, green)

**Fig. 3 | AAV6-CXCL9 directed lymphocyte chemotaxis. a** Diagrammatic overview of in vitro competitive T lymphocyte chemotaxis assay. **b** Lymphocyte (CTV + , blue) chemotaxis in AAV6-EGFP (control, green) transduced or AAV6-CXCL9 (RFP + , red) transduced GL261 field at 1- and 24 h following co-culture. Statistical analyses performed by two-way ANOVA with Sidak's multiple comparisons test, $n = 3$ per time point, per group. Representative images shown. Dashed white line represents the lymphocyte-tumor border at assay start. **c** Competitive chemotaxis measured as described in (**b**) in C8-D1A astrocytes field at 1- and 24 h following co-culture. **d** Schematic outlining combination AAV6 and PD-1 ICB treatment and tissue collection and survival analysis. Flow cytometric detection of tumor-infiltrating CD8+ lymphocytes in control, single or combination AAV6-CXCL9 plus anti-PD-1 ICB treatment in (**e**) GL261 and (**f**) KR158 models. Fold-change normalization based on sham values (dashed black line). Statistical analysis performed using ordinary one-way ANOVA with Fisher's least significant difference (LSD) test for multiple comparisons, $n = 3-6$ per treatment group, individual values shown. Flow cytometric detection of tumor-infiltrating CD4+ lymphocytes in control, single or combination AAV6-CXCL9 plus anti-PD-1 ICB in (**g**) GL261 and (**h**) KR158 models. Fold-change normalization based on sham values (dashed black line).

Statistical analysis performed using ordinary one-way ANOVA with Fisher's LSD test for multiple comparisons, $n = 3-6$ per treatment group, individual values shown. **i** 3D IHC of lymphocytes (EGFP + , green) in sham tumor control (scale bar 100 μm) and combination AAV6-CXCL9 plus anti-PD-1 ICB treated GL261 tumors (scale bar 70 μm) isolated day 15. Tissues counterstained with DAPI (blue) and GFAP (astrocytes, red). Images representative of $n = 3$ per treatment group. **j** Representative 3D IHC of a region of lymphocyte infiltration (EGFP + , green) in combination treated GL261 tumors, $n = 3$, scale bar 30 μm. Tissues were counterstained with DAPI nuclear dye (blue) and GFAP (astrocytes, red). Voxel-based co-localization between EGFP and GFAP indicate areas of convergence, 'cell junctions', between astrocytes and T cells, shown as a separate channel (white). 3D surface renderings of astrocytes and lymphocytes to visualize cell junctions in greater detail. P-values ≤ 0.05 are considered statistically significant. Bar graphs depict group mean with error bars representing standard deviation. Box-whisker plots display the box ranging from the first to the third quartile, the center median value, and the whiskers extend from each quartile to the minimum and maximum values. Source data are provided as a Source Data File.

(Fig. 3j). While cellular communication between these two cell populations can be inferred given the physical proximity and direct contact, the exact nature or respective influence imparted by these interactions remain an area of ongoing research.

### AAV6-CXCL9 sensitizes preclinical GBM to anti-PD-1 ICB
To assess if enhanced lymphocyte recruitment and immunological reprogramming through combination treatment could produce anti-tumor responses against GBM, we performed survival analyses in both the GL261 and KR158 syngeneic model systems. Five days following tumor implantation, AAV6 encoding CXCL9 or EGFP control transgene was injected intratumorally, with anti-PD-1 ICB (10 mg/kg) administered intraperitoneally for a total of 4 doses given every 72 h (Fig. 3d). In the GL261 model we found that anti-PD-1 ICB produced a small, but non-significant increase in overall survival as compared to sham treated control animals ($p = 0.060$), where AAV6-CXCL9 treatment yielded no survival benefit as a monotherapy (Fig. 4a). Combination treatment significantly improved overall survival, with 50% of animals exhibiting durable outcomes (Fig. 4a). We observed similar results in the KR158 model, which carries a low mutational burden and is recalcitrant to immunotherapy, with combination treatment significantly improving median survival, and long-term survival observed in 25% of this cohort (Fig. 4b). As an additional metric to validate the ability of combination AAV6-CXCL9 plus anti-PD-1 ICB to immunologically transform GBM tumors, GL261 tumors were implanted in GREAT transgenic mice to evaluate tumor-wide IFNγ expression following treatment as described in Fig. 3d. IFNγ was readily detected in combination treated tumors as compared to sham control treated tumors, evidenced by EYFP signal detection via 3D IHC (Fig. 4c). Immunolabeling of tissues for CD45 confirms that EYFP (IFNγ)+ cells are immune cells (Fig. 4d), indicative of pro-inflammatory immune activation.

To determine if CD8 lymphocytes contribute to the therapeutic survival effect, we repeated combinatorial treatment with concomitant CD8 depletion (Fig. 4e) in the GL261 model. We found that on study day 18 all animals treated with CD8 depleting antibodies had no detectable levels of circulating CD8 T lymphocytes, and no changes in the quantity of circulating CD4 T lymphocytes (Fig. 4f, Supplementary Fig. S5d). CD8 depletion reversed the survival benefit observed with combination AAV6-CXCL9 plus anti-PD-1 ICB, and this cohort progressed as quickly as control treated subjects (Fig. 4f). We similarly evaluated the impact of CD4 lymphocyte depletion (Fig. 4e, Supplementary Fig. S5e), and found that while there was a trend for reduced survival in animals treated with combination therapy plus concomitant CD4 depletion, these results were not considered to be statistically significant, and many animals in the cohort demonstrated extended survival as compared to control (Fig. 4g). Interestingly, sham control

animals with concomitant CD4 depletion demonstrated more rapid disease progression as compared to non-depleted controls, indicating an important role for these lymphocytes in host tumor response (Fig. 4g). To determine if combination treatment could confer long-term immune memory formation, we performed a GL261 tumor rechallenge in long-term survivors (>55 days) that had received AAV6-CXCL9 plus anti-PD-1 ICB. No observable residual tumor was present from the initial tumor implantation during the second implantation. A second cohort of age-matched naïve animals was intracranially injected with GL261 as a control. Control animals all succumbed to tumor burden within 30 days of tumor implantation, whereas 100% of rechallenge animals remained disease free (Fig. 4h). These data confirm that therapeutic response to combination therapy is dependent on tumor infiltration by CD8 T lymphocytes as part of the adaptive immune cascade, and combination therapy can convey long-term immune memory protection against recurrence.

To assess if intratumor delivery of AAV6-CXCL9 is vital for therapeutic efficacy we compared overall survival in GL261 tumor bearing mice where AAV6-CXCL9 (1e10 VGS) was intratumorally (IT) injected, or injected into the contralateral (CL) hemisphere at coordinates symmetrical to intratumor delivery. We found that intratumor delivery of AAV6-CXCL9 produced a more robust survival response as compared to contralateral delivery (Supplementary Fig. S5f), although a subset of these animals did demonstrate extended survival as compared to control. We also tested therapeutic efficacy in more established tumors, comparing AAV6-CXCL9 intratumor delivery plus combination anti-PD-1 ICB given on day 5 or day 12 (delayed treatment, DT). While delayed treatment yielded some survival benefit as compared to control treated animals, is was significantly reduced as compared to animals receiving combination treatment beginning on day 5 (Supplementary Fig. S5g). Whole brain imaging of GL261 tumors excised 1 week following intratumor delivery at day 12 with AAV6 encoding an mCherry fluorescent reporter transgene revealed extensive local and distant disease progression, including ventricular dissemination (Supplementary Fig. S5h). AAV6 transduction in these larger tumors was confined largely to the site of intratumor injection (Supplementary Fig. S5h), indicating that in larger tumors a multifocal delivery method may be required for optimal anti-tumor efficacy.

### scRNAseq identifies treatment-related immune response to AAV6-CXCL9 and anti-PD-1 ICB
In an effort to define the immunological landscape of AAV6-CXCL9 treated tumors with or without concurrent anti-PD-1 ICB, we performed single cell RNA sequencing (scRNAseq) on CD45-positive cells isolated from GL261 tumors collected on day 15 of treatment as outlined in Fig. 3d. Top differentially expressed genes from each pooled

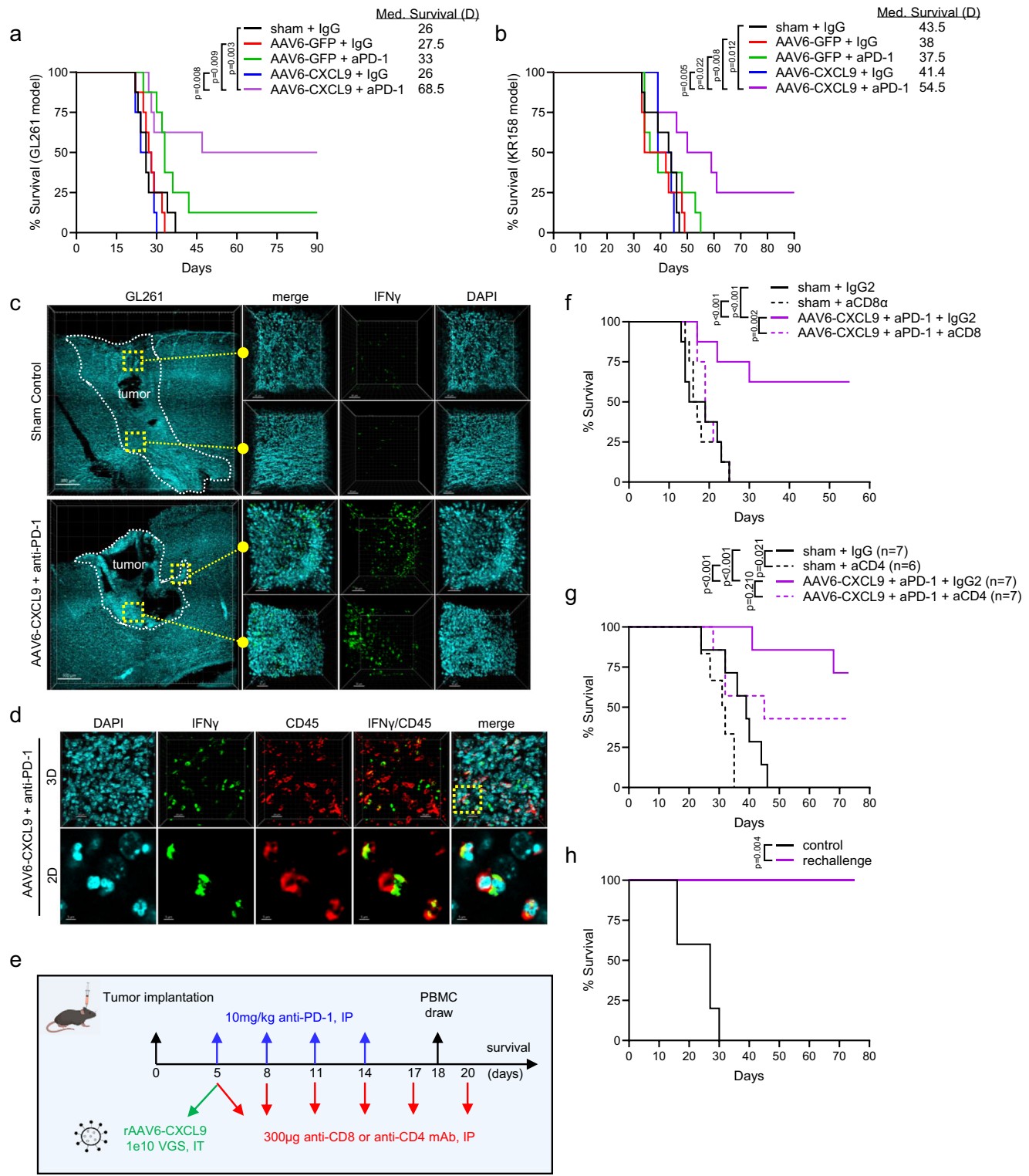

population were identified, and cluster cell types were defined using the expression of known marker genes resulting in the identification of 13 unique cell clusters[34,35] (Fig. 5a, Supplementary Table 1). Dimensionality reduction using uniform manifold approximation and projection (UMAP) was performed on 52,344 cells collected across five treatment groups: sham (saline), AAV6-ctrl + IgG, AAV6-ctrl + aPD-1, AAV6-CXCL9 + IgG, and combination AAV6-CXCL9 + aPD-1, 3 mice per group (Fig. 5b, c, Supplementary Fig. S7a–d). Analyses of lymphocyte tumor recruitment across treatment groups recapitulate our earlier observation, with combination therapy yielding a significant increase

in total infiltrating CD8 T lymphocytes (Fig. 5d), identified using the gene expression markers *Cd3d, Cd8a, Cd8b1* as previously described[36]. T regulatory lymphocytes (Treg), defined by *Cd4*, *Foxp3*, and *Il2ra* gene expression[36], were also increased in response to combination therapy, although collectively these represent <1% of the total tumor-associated immune population (Fig. 5e). Increased tumor infiltration by monocytes, classified by high *Ly6c1* expression, was observed across all treatment groups as compared to sham control mice (Fig. 5f), with an enrichment of non-classical monocytes characterized by *Spn, Cx3cr1, and Tnfrsf1b* expression[37] in groups receiving anti-PD-1 treatment

**Fig. 4 | AAV6-CXCL9 sensitizes GBM tumors to anti-PD-1 immunotherapy.**
Survival analysis in (**a**) GL261 and (**b**) KR158 tumor-bearing mice treated with control, AAV6-CXCL9, and anti-PD-1 ICB alone and in combination. Median survival for each treatment group shown, $n = 8$ per group. Statistical analysis was performed using Log-rank (Mantel-Cox) test comparing individual treatment groups. **c** Tile-stitch 10x 3D IHC imaging of GL261 tumors resected from sham control and combination AAV6-CXCL9 plus anti-PD-1 ICB treated GREAT mice, $n = 3$ per group, scale bar 300 μm. DAPI nuclear dye (blue) used to identify tumor area outlined by the dashed white line. Digital magnification of regions outlined in the far-left panel to show higher image resolution for each treatment, scale bar 50 μm. EYFP (green) correlates with IFNγ expression. **d** 3D IF of tissue from GL261 tumor tissue as shown in (**c**) immunolabeled for CD45 expression (red), scale bar 20 μm. Digital zoom of region outlined in the far-right panel shows co-localization between CD45 and IFNγ, indicating these are immune cells, $n = 3$, scale bar 5 μm. **e** Diagrammatic summary of combination treatment strategy with concomitant CD4 or CD8α antibody depletion. **f** Survival analysis in GL261 tumor-bearing mice treated with combination AAV6-CXCL9 plus anti-PD-1 monoclonal antibody, with or without anti-CD8α depletion. Statistical analysis performed using Log-rank (Mantel-Cox) test comparing individual treatment groups, $n = 8$ per group. **g** Survival analysis in GL261 tumor-bearing mice treated with combination AAV6-CXCL9 plus anti-PD-1 monoclonal antibody, with or without anti-CD4 depletion. Statistical analysis performed using Log-rank (Mantel-Cox) test comparing individual treatment groups, $n = 7-8$ per group. **h** Survival analysis in long-term survivors from combination treated animals re-challenged with tumor at day 55 of study ($n = 5$). Age-matched naïve control mice were orthotopically implanted with GL261 as survival control arm. Statistical analysis was performed using Log-rank (Mantel-Cox) test comparing individual treatment groups. *P*-values ≤ 0.05 are considered statistically significant. Source data are provided as a Source Data File.

(Fig. 5g). Graphical summaries for all remaining cells clusters in response to each treatment are shown in Supplementary Fig. S6a–h. Given that CXCR3 is the concomitant receptor for CXCL9, we leveraged our scRNAseq data to assess which tumor-associated immune cells could interact with our AAV encoded transgene. We found that *Cxcr3* was principally expressed by lymphocytes, as shown on the UMAP in Fig. 5h. Further classification of lymphocyte subsets was done to determine if specific populations would be more or less responsive to AAV-CXCL9 therapy by means of differential CXCR3 expression. We identified 11 unique lymphocyte clusters (Fig. 5i, Supplementary Fig. S7e), with equal distribution of *Cxcr3* transcript expression detected across all clusters, with cluster 9 and 10 exhibiting slightly reduced overall *Cxcr3* transcript expression (Fig. 5i, Supplementary Fig. S7e). These findings suggest that CXCL9 broadly interacts with all lymphocyte subsets through CXCR3 expression.

## Combination AAV6-CXCL9 and anti-PD-1 ICB treatment increases cellular crosstalk in lymphocytes

As shown in Fig. 6a, we identified 2,260 differentially expressed genes (DEGs) associated with AAV6-CXCL9 treatment, 2,607 DEGs associated with anti-PD-1 treatment, and 2,649 DEGs associated with these treatments combined. Of these, 70, 194, and 151 DEGs appear to be unique to each given treatment strategy, respectively, and may provide unique insight toward treatment impact on immune cell functional states. Through transcriptional expression of distinct ligands and receptors, cell-type-specific interactions were inferred, providing additional insight towards the inflammatory profile of tumors and how they change in response to treatment[38]. Using our predefined cell clusters, a simplified DEG set was established for each. DEGs were then queried against public ligand-receptor databases (see Methods). Summary results are shown in Chord Plots, where line thickness represents the number of predicted interactions between two defined cell clusters (Supplementary Fig. S8a–e). Next, we performed direct comparisons of interactome activity between treatment groups to elucidate heightened or decreased connectivity associated with AAV6-CXCL9 and anti-PD-1 ICB, where heatmap relative values in red indicate increases and blue decreases in prospective ligand-receptor interactions. In evaluating AAV6-CXCL9 in combination with either anti-PD-1 ICB or IgG2 control to resolve the contributions of ICB, notable increases in signals emanating from each macrophages (Mac), border-associated macrophages (BAM), microglia (Mg), and NK cells signaling to CD8+ and regulatory T cells were observed (Fig. 6b). Decreased incoming signals were noted in BAMs, CD4 + T cells, and dendritic cells (DCs) stemming from nearly all cell clusters (Fig. 6b). Cell-cell interactions associated with AAV6-CXCL9 shown in Fig. 6c reveal heightened communication directed toward both CD4+ and CD8 + T cell subsets, and NK cells prompted by all clusters excluding B cells and DCs. Signaling originating from all lymphocyte populations, and most innate immune cells including Macs, Mgs, Monocytes, and NK cells was

increased, suggesting that AAV6-CXCL9 treatment broadly stimulates immune activity.

Given that combination treatment promotes CD8 T cell tumor infiltration, which is required for anti-tumor efficacy, we sought to resolve how treatment might impact CD8 T cell effector function via pathway analysis of DEGs specifically within these cells. Comparative pathway analysis between CD8 T cell DEGs shows selective enrichment of thrombospondin (*Thbs1*), poliovirus receptor (*Pvr*, *Cd155*), *Cd137* (*Tnfrsf9*, *4-1BB*), fibronectin-1 (*Fn1*), laminin (*Lamc1*), and several genes associated with major histocompatibility complex class I signaling, among others, as uniquely affiliated with combination therapy when compared to AAV6-CXCL9 plus IgG2 control (Fig. 6d). These data suggest that anti-PD-1 treatment prompts T cell activation via CD137[39], but also reciprocal immune suppression via CD155 given its inhibitory function as a ligand for T cell immunoreceptor with immunoglobulin and ITIM domain (TIGIT)[40]. FN1, laminin, and THBS are major constituents of the extracellular matrix, and when produced by lymphocytes have been described to support cell-cell engagement, transendothelial migration, and lymphoproliferation[41–43]. DEG comparisons between combination therapy and AAV6-EGFP control plus anti-PD-1 ICB treatment reveals selective pathway enrichment of macrophage migration inhibitory factor (*Mif*), growth arrest specific (*Gas6*), galectin-9 (*Lgals9*), inducible T cell co-stimulator (*Icos1*), tumor necrosis factor (*Tnf*), and pleiotrophin (*Ptn*) as a result of AAV6-CXCL9 treatment (Fig. 6e). These data infer that AAV6-CXCL9 directly promotes CD8 T cell activation through increased ICOS and TNF expression[44,45], T cell migration via PTN[46], and reciprocally enhances innate immune stimulation of NK cells and myeloid cells via GAS6 and MIF secretion, respectively[47,48]. It also reveals galectin-9 as a possible mechanism for CD8 T cell acquired exhaustion[49,50]. Both anti-PD-1 ICB and AAV6-CXCL9 treatments stimulate NOTCH, TGFβ, IL-10, SEMA4, CXCL, Complement, and CCL pathway activation (Fig. 6d, e), each with varying impact on T cell maturation, effector function, homeostasis, survival, and migration[51–57]. Comparative pathway analysis was also performed for CD4 T cells across treatment groups (Supplementary Fig. S8f, g).

We next leveraged the NanoString nCounter® Immune Exhaustion Panel to further characterize immune status and inflammatory signatures associated with each respective treatment. A summary of pathway activation across all cell subsets in response to individual treatments is shown in Fig. 6f and Supplementary Fig. S8h, with CD8 T cell clusters outlined in black for each treatment group. From the NanoString analysis, CD8 T cell specific differential pathway activation is graphically presented in Fig. 6g–l, where we identify treatment-associated changes in each antigen presentation (Fig. 6g), chemokine signaling (Fig. 6h), cytotoxicity (Fig. 6i), T cell exhaustion (Fig. 6j), TCR signaling (Fig. 6k), and PD-1 signaling (Fig. 6l). In particular, combination AAV6-CXCL9 plus anti-PD-1 ICB was associated with the highest increase in cytotoxicity, TCR signaling, and PD-1 signaling. Increased T

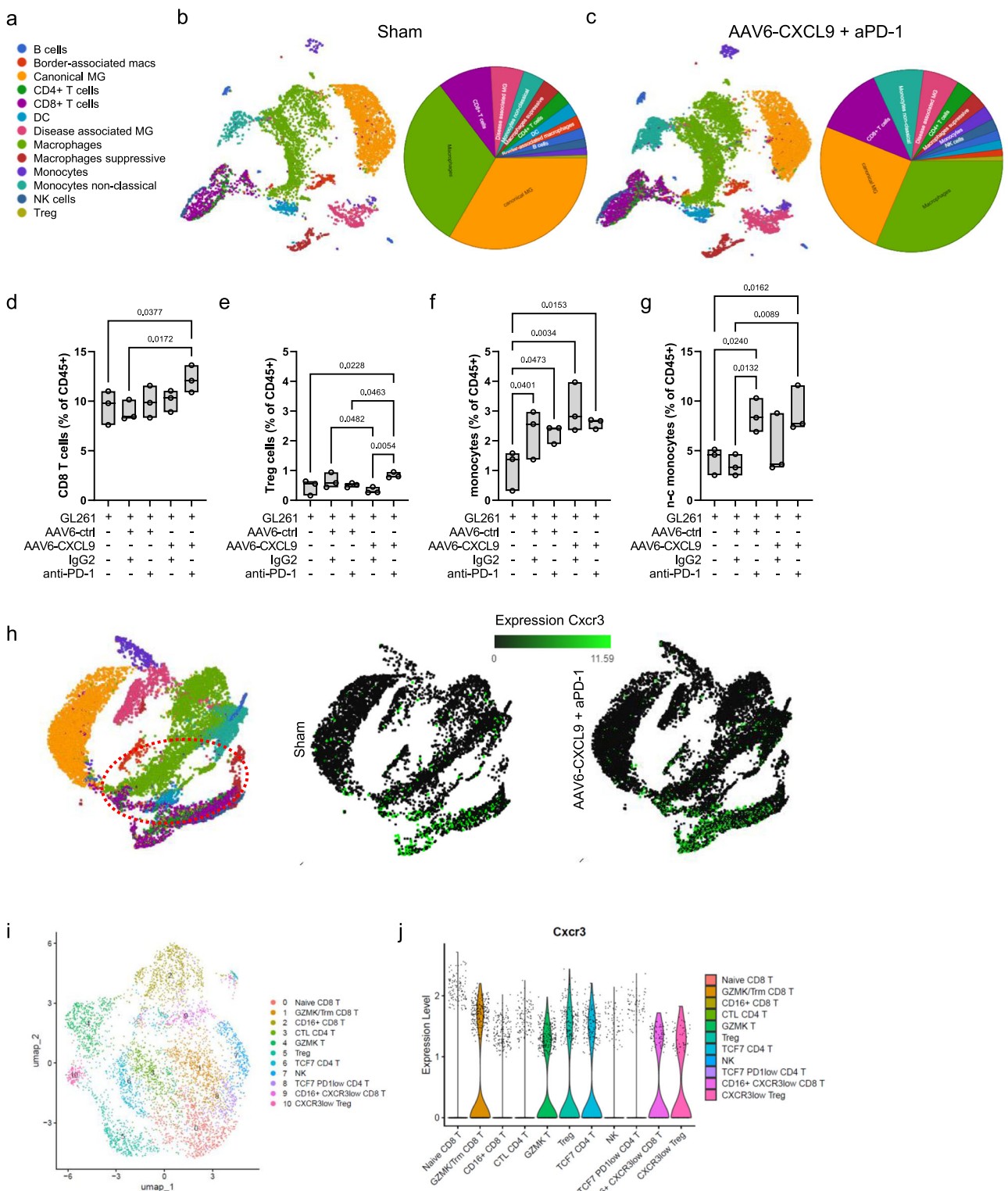

**Fig. 5 | Immunological landscape of GBM tumors treated with AAV6-CXCL9 and anti-PD-1 immunotherapy. a** Summary of UMAP cell clusters. UMAP of cell types clustered by scRNA transcriptional analysis of 52,344 CD45+ cells isolated from GL261 tumor bearing mice treated with (**b**) sham (saline) or (**c**) combination AAV6-CXCL9 + aPD-1 treated GL261 tumors, *n* = 3 mice per group. Summary circle chart depicting cell cluster population frequency detected for each treatment included alongside each UMAP. Quantitative change in population frequency of (**d**) CD8 + T cells, (**e**) Treg cells, (**f**) monocytes, and (**g**) non-classical (n−c) monocytes across

treatment groups. Statistical analyses performed using ordinary one-way ANOVA with Fisher's LSD test for multiple comparisons, *n* = 3 per group, individual values shown. **h** UMAP projection of *Cxcr3* transcript expression (TPM) detected across all cell clusters. **i** UMAP of T cell clusters. **j** Summary of *Cxcr3* transcript expression across defined T cell clusters. Box-whisker plots display the box ranging from the first to the third quartile, the center median value, and the whiskers extend from each quartile to the minimum and maximum values. Source data are provided as a Source Data File.

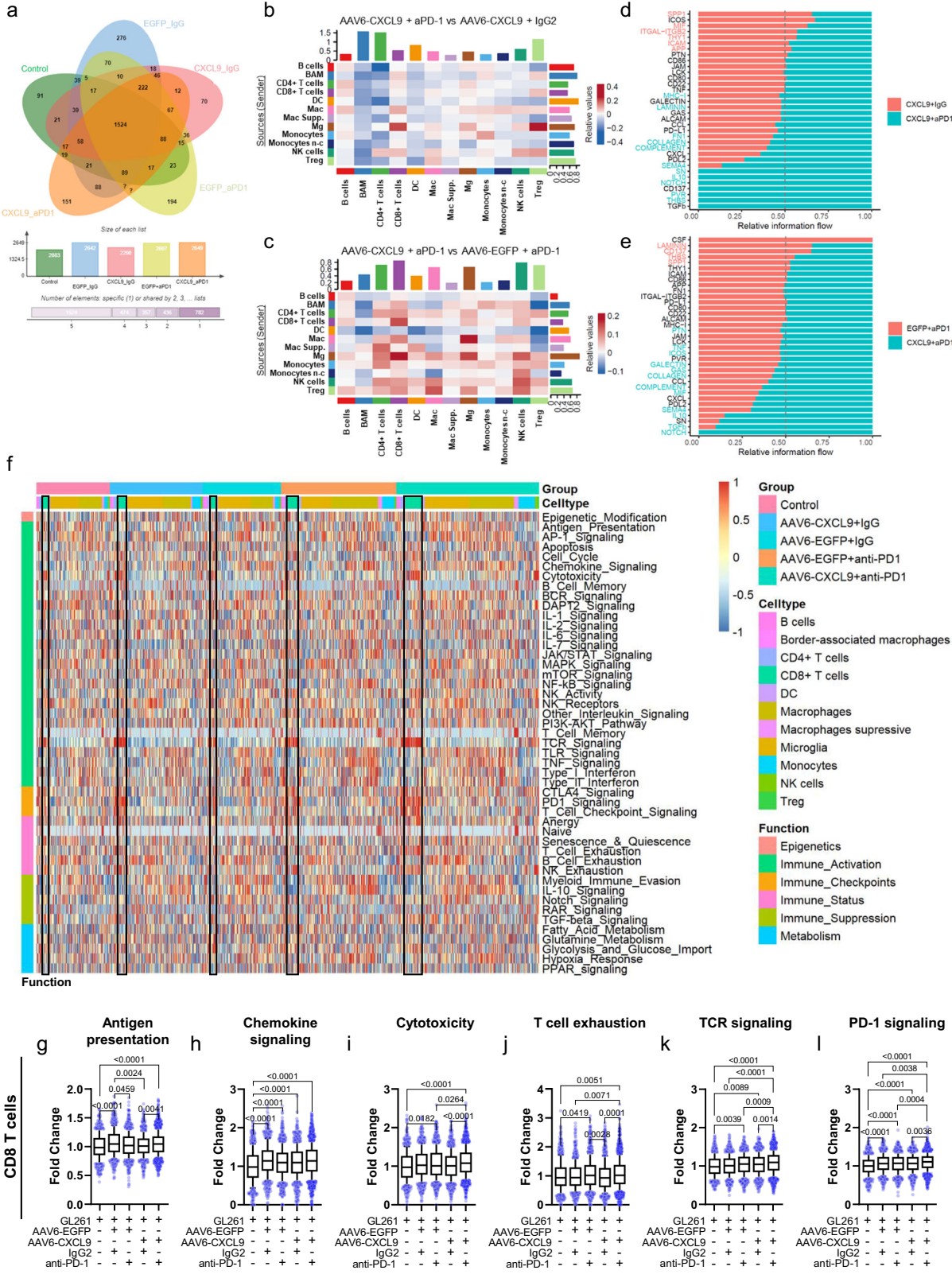

cell exhaustion appears to be associated with AAV6-CXCL9 treatment. Given that monocyte tumor infiltration was additionally increased in response to treatment (Fig. 5f, g), we evaluated pathway activation in these cells to better understand their functional status. We found enhanced activation across 12 pathways, including antigen presentation, chemokine signaling, cytotoxicity, IL-10 signaling, JAK/STAT signaling, other interleukin signaling, T cell checkpoint, TGFβ signaling,

TNF signaling, Type I interferon signaling, and Type II interferon signaling (Supplementary Fig. S9). Of these, AAV6-CXCL9 treatment appears to be associated with increased antigen presentation, cytotoxicity, JAK/STAT signaling, and Type I interferon signaling, where anti-PD-1 ICB induces IL-10 signaling, TLR signaling, and TNF signaling. Together these data are suggestive that treatment may augment the pro-inflammatory function of these cells.

**Fig. 6 | AAV6-CXCL9 and anti-PD-1 immunotherapy stimulates CD8 lymphocyte activation. a** Venn Diagram representing differentially expressed genes affiliated with each treatment. **b** Heatmap depicting scRNA-seq-derived cell-cell communication networks enriched or decreased in response to combination AAV6-CXCL9 + aPD-1 as compared to AAV6-CXCL9 + IgG2 treatment across identified cell clusters. **c** Heatmap depicting scRNA-seq-derived cell-cell communication networks enriched or decreased in response to combination AAV6-CXCL9 + aPD-1 as compared to AAV6-EGFP + aPD-1 treatment across identified cell clusters. **d** Waterfall summary plot of scRNA-seq-derived signaling pathways enriched in CD8 + T cells following combination AAV6-CXCL9 + aPD-1 as compared to AAV6-CXCL9 + IgG2 treatment. **e** Waterfall summary plot of scRNA-seq-derived signaling pathways enriched in CD8 + T cells following combination AAV6-CXCL9 + aPD-1 as compared to AAV6-EGFP + aPD-1 treatment. **f** Heatmap representation of gene

expression analysis derived from all cell clusters using the nCounter® Immune Exhaustion Panel (nanoString) following AAV6-CXCL9 gene therapy with or without PD-1 ICB. CD8 + T cell populations outlined in black for each treatment group. **g–l** Quantification of common pathways found to be differentially regulated in CD8 + T cells in response to treatment, derived from n = 669, 674, 1099, 912, and 1656 single cells pooled from n = 3 individual samples per treatment group, graphically presented from left to right. Statistical analyses performed using Kruskal–Wallis test followed by Dunn's multiple comparisons, with individual values shown. *P*-values ≤ 0.05 are considered statistically significant. Box-whisker plots display the box ranging from the first to the third quartile, the center median value, and the whiskers extend from each quartile to the minimum and maximum values. Source data are provided as a Source Data File.

## Cytokine profiling of combination AAV6-CXCL9 plus anti-PD-1 ICB

As combination therapy increases DEGs of both the CCL and CXC superfamily of secreted chemokines and cytokines, we sought to parse out transcriptional changes within CD8 T cells as an additional means to evaluate the activation state of these cells given the central role of these ligands in directing migration and activation of immune cells during inflammation[58]. A summary of all CCL and CXC family ligand and receptor transcripts expressed by CD8 T cells is presented in the heatmap in Fig. 7a. CD8 T cell mediated stimulation of monocytes/macrophages is evidenced by increased transcription of *Ccl2*, *Ccl3*, and *Ccl12* across all treatment groups as compared to sham control (Fig. 7b–d). *Ccl4* transcription was also increased (Fig. 7e), indicative of NK stimulation by CD8 T cells. *Ccl5* was found to be the most differentially upregulated soluble ligand in combination treated CD8 T cells as compared to all other treatment groups (Fig. 7f), and is indicative of CD8 T cell effector function[59,60]. While we show that AAV6 delivered CXCL9 transgene expression predominantly emanates from tumor-reactive astrocytes, our scRNAseq data shows that each anti-PD-1 and AAV gene therapy induces CXCL9 transcription within CD8 T cells (Fig. 7g), additionally demonstrating immune activation as a result of treatment[22,23,61]. *Cxcl10* was also found to be transcriptionally upregulated in response to anti-PD-1 and AAV gene therapy (Fig. 7h), which prompts further CD4, CD8, and NKT lymphocyte recruitment[61]. Altogether, these data support that combination AAV6-CXCL9 and anti-PD-1 ICB both increases lymphocyte trafficking to intracranial GBM tumors and potently stimulates effector lymphocyte cellular communication and activation.

As described above, secreted cytokines can influence the trajectory of tumors in a multitude of ways- reprogramming tumor-associated cells and suppressing infiltrating inflammatory subsets which allows for tumor tolerance, progression, metastasis, and even therapeutic resistance or, alternatively, creating an environment favorable for innate and adaptive immune activation to facilitate tumor rejection[62]. Moreover, the cytokine profile of a tumor may serve as predictive and/or therapeutic biomarkers allowing for the detection of tumor presence, forecasting therapeutic response, and can also be used to guide therapeutic choices[62]. We performed a large-scale cytokine proteomic assessment of single agent and combination treated tumors to identify candidate biomarkers of response to therapy. Tumors were resected 10 days after the onset of treatment as shown in Fig. 3d. Of the 111 soluble murine proteins on the array, relative expression of 65 was detected in treated and/or control GL261 tumor samples as summarized Supplementary Fig. S10a, with representative cytokine immunoblots shown in Fig. 7j. 10 secreted factors were identified as differentially expressed as compared to sham control tumors following either single or combination treatment with AAV6-CXCL9 and anti-PD-1 ICB: ADIPOQ, C1QR1 (CD93), CCL5, CCL12, CD40, CXCL9, CXCL10, CXCL16, LCN2, and MPO (Fig. 7k, Supplementary Fig. S10b–k). Of these, CCL5, CD40, and CXCL16 were most potently induced by combination treatment. These markers are

indicative of lymphocyte presence and activation, where CCL5 is a potent pro-inflammatory ligand manufactured principally by CD8 T lymphocytes, and CD40 is a co-stimulatory ligand that triggers lymphocyte proliferation and cytokine production[59,60]. Of note, elevated CCL5 ligand expression demonstrates concordance with scRNAseq data (Fig. 7f). In addition, CXCL9, CXCL10 and CXCL16 are strong chemotactic signals for lymphocyte recruitment. Both CXCL10 and CXCL16 are induced by interferon gamma (IFNγ) and tumor necrosis alpha (TNFα), powerful catalysts of innate and adaptive inflammation[63,64]. A summary of treatment-induced secreted ligands and known receptor interactions are depicted via circular interactome analysis performed using Circos® visualization software[65], revealing insight towards immune reprogramming that occurs in response to each respective treatment (Fig. 7l). These data combined validate that AAV6 delivery of CXCL9 to the tumor microenvironment in tandem with anti-PD-1 ICB not only facilitate lymphocyte recruitment to GBM tumors, but also reprograms the immunological landscape towards a pro-inflammatory phenotype.

In summation (Fig. 8), intra-tumor delivery of AAV6 encoded CXCL9 results in the production of a pro-lymphocyte chemotactic gradient by transduced tumor-reactive astrocytes. This, in concert with anti-PD-1 ICB, significantly increases tumor infiltration by lymphocytes likely through CXCL9 engagement with its cognate receptor expressed by these cells- CXC motif chemokine receptor 3 (CXCR3). In particular, CD8 T lymphocytes are the premier arbiters of anti-tumor response, where depletion of this lymphocyte subset negates therapeutic efficacy. Moreover, CD8 T cell effector activation and function is evidenced by heightened expression of co-stimulatory molecules, such as 4-1BB and ICOS, and production of pro-inflammatory chemokines and cytokines. Beyond CD8 T lymphocyte activation, AAV6-CXCL9 and anti-PD-1 ICB appear to contribute widespread immunological activation, demonstrated by heightened cellular cross-talk across numerous immune clusters, and protein detection of pro-inflammatory molecules. Notably, CCL5, CXCL9, CXCL10, and CD40 are detected in response to combination therapy within resected tumors, and may serve as biomarkers of therapeutic response.

## Discussion

Perhaps one of the most consequential advantages of AAV gene therapy for the treatment of GBM, and possibly other solid tumors, is that fundamentally AAV is a modality for distributing encoded transgene into the tumor microenvironment. Still further work has been done on modification and design of vectors carrying unique biotherapeutic transgenes capable of targeting particular cells. These can be pooled, offering a straightforward method for delivering personalized anti-cancer combination treatment targeting one or multiple aspects of tumorigenicity. Examples include transgenes encoding anti-angiogenics, anti-migrastics, direct cytotoxic agents (e.g. suicide genes), immune stimulating elements, immune checkpoint inhibitor decoys, and even gene-editing elements such as CRISPR/Cas9 or shRNA. Herein, we demonstrate that AAV encoding for the lymphotactic

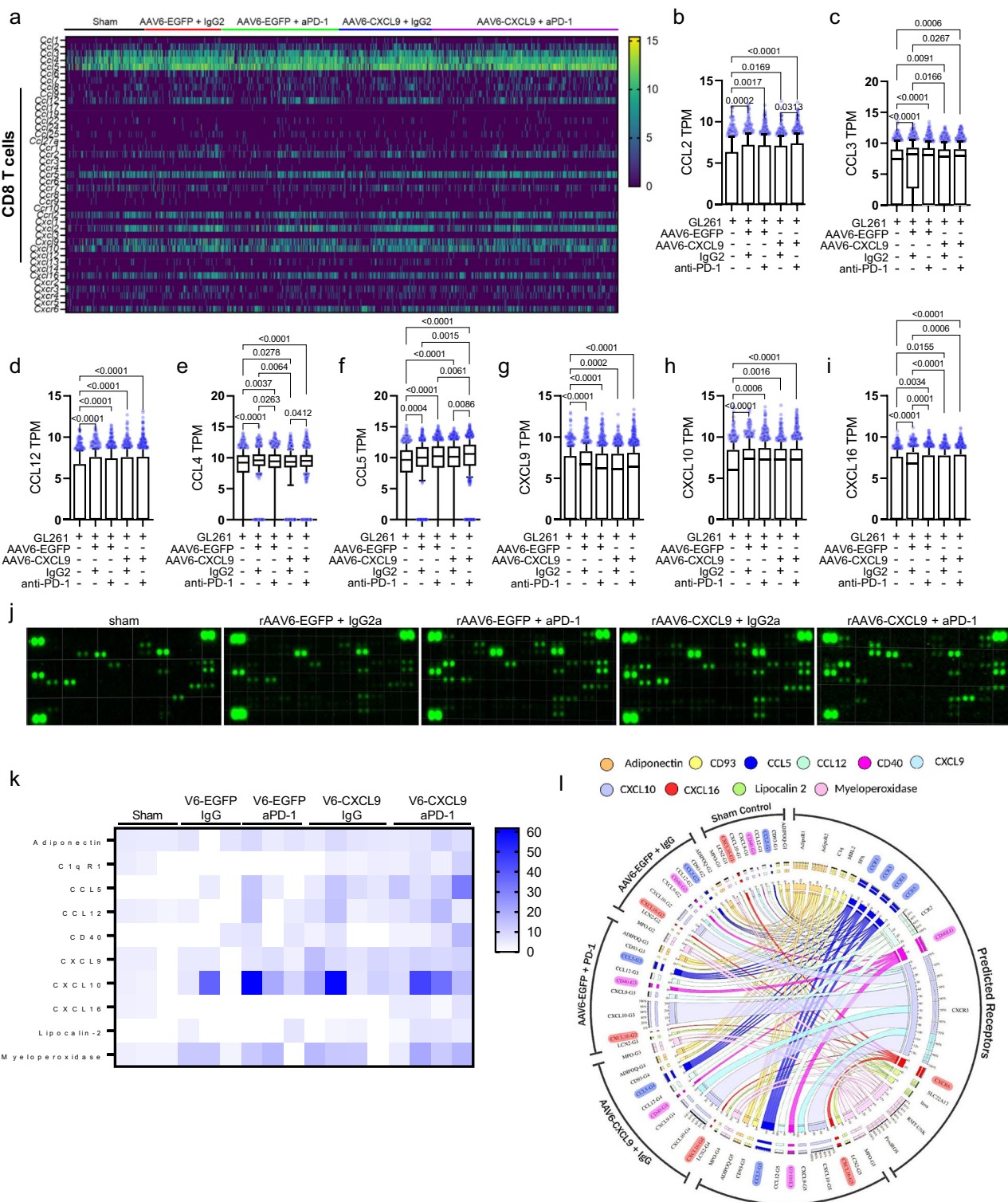

**Fig. 7 | Inflammatory signature of preclinical GBM treated with AAV6-CXCL9 and anti-PD-1 ICB. a** Heatmap summary of scRNA-seq-derived CCL-CXC expression in CD8 + T cells isolated from GL261 tumors in response to AAV6-CXCL9 and anti-PD-1 ICB treatment created using GraphPad Prism. **b–i** Quantification of CCL-CXC genes found to be differentially expressed in CD8 + T cells in response to treatment derived from *n* = 669, 674, 1099, 912, and 1656 single cells pooled from *n* = 3 individual samples per treatment group, graphically presented from left to right. Statistical analyses performed using Kruskal–Wallis test followed by Dunn's multiple comparisons, with individual values shown. **j** Representative immunoblots depicting chemokine and cytokine protein expression detected in GL261 tumors

resected following treatment with AAV6-CXCL9 with and without PD-1 ICB (*n* = 3 for sham, rAAV6-EGFP + IgG2a, and rAAV6-EGFP + aPD-1; *n* = 4 for rAAV6-CXCL9 + IgG2a and rAAV6-CXCL9 + aPD-1). -4 per group. **k** Heatmap summary of CCL-CXC relative protein expression found to be differentially expressed in response to AAV6-CXCL9 with and without PD-1 ICB, created using GraphPad Prism. **l** Circos interactome analysis of detected differentially expressed proteins and predicted receptors. *P*-values ≤ 0.05 are considered statistically significant. Box-whisker plots display the box ranging from the first to the third quartile, the center median value, and the whiskers extend from each quartile to the minimum and maximum values. Source data are provided as a Source Data File.

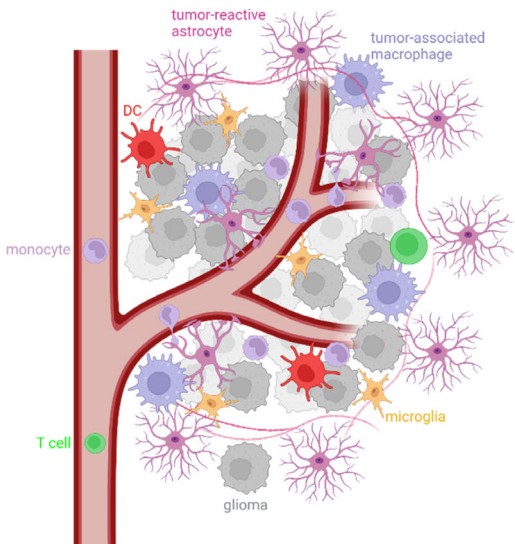
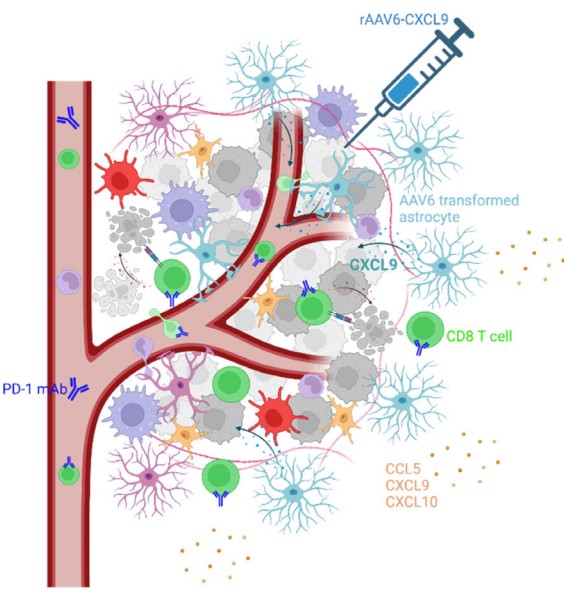

**Fig. 8 | Diagrammatic summary of findings.** Intra-tumor delivery of AAV6 encoding CXCL9 results in robust transduction of tumor-reactive astrocytes, creating a chemotactic gradient of secreted CXCL9. This improves lymphocyte trafficking in combination with anti-PD-1 ICB through chemokine-receptor engagement between CXCL9 in the TME and CXCR3 expression on lymphocytes.

CD8 + T cells are required for durable survival response to treatment, indicating that tumor cell killing is mediated by the adaptive arm of immunity. Combination treatment also transforms the inflammatory milieu of tumors, creating a pro-inflammatory environment evidenced by the presence of cytokines and chemokines that further promote innate and adaptive immune activation.

chemokine CXCL9 can be leveraged to engage the immune system to recognize and attack tumor cells by modulating anti-PD-1 immunotherapy in the combinatorial setting.

We found that GBM tumors possess a chemokine signature that favors the recruitment of myeloid and other 'suppressive' immune cells, notably lacking lymphocyte call-and-receive signals. This is unsurprising given that the immune contexture of human GBM is largely comprised of myeloid cells[66]. To test if GBM reconstitution with a lymphocyte selective chemokine could improve trafficking we leveraged AAV gene therapy to generate durable production of CXCL9. With the initial intent of transducing glioma cells to generate tumor-tropic transgene expression, we found that in vivo transduction was not redolent of in vitro screening methods, instead revealing potent transduction of tumor-reactive astrocytes with our lead serotype, AAV6. These findings advise caution in extrapolating AAV transduction efficacy from in vitro or even ex vivo screening methods to anticipated cell/tissue tropism in the in vivo setting. Despite this discrepancy, we found that AAV6-transduced astrocytes confer a high degree of tumor tropism, where transgene expression was limited to the immediate tumor area. Whether this is due to localization of AAV delivery at the injection site, which is an advantage of direct inoculation over systemic administration, or through changes to the reactive astrocytes surrounding the tumor that lead to increased transfection susceptibility is an area of active investigation. Our study additionally shows that AAV6 inoculation in vivo in non-tumor bearing mice preferentially transfects astrocytes, potentially indicating the inoculation itself is an injury/inflammatory event that sequesters the therapy at the site of delivery. Altogether our findings suggest that astrocytes may be selectively susceptible to AAV6 transduction as compared to the other cells constituting the tumor microenvironment that reside within the treatment field of the viral administration. Heparin sulfate proteoglycan (HSPG), one of the principal host cell receptors for AAV6 transduction[67] is reported to be upregulated by astrocytes in response to brain injury[30,31], offering a possible explanation for the tumor-tropic nature of astrocyte transduction by AAV6.

AAV targeting of tumor-associated cells may carry several distinct advantages over directly targeting cancer cells. Given the genetic diversity of cancer cells, identifying a serotype that can reliably and consistently transduce tumor cells poses a challenge. Tumor-associated cells such as astrocytes, as demonstrated herein, or alternatively endothelial cells, microglia, etc. are more likely to have a homogenous genotype across patient tumors allowing for targeted off-the-shelf therapeutic development, drastically accelerating treatment timelines and reducing cost as compared to personalized medicine approaches necessary to guide serotype selection for cancer cell targeting. Our data also indicates that targeting of tumor-associated astrocytes is likely to produce a more durable response as these cells are less susceptible to genetic alterations. Direct tumor transduction, on the other hand, is short-lived, likely as a result of vector genome dilution due to the high proliferative capacity of these cells[68] and may require repeat treatments to sustain transgene expression. This is supported by other studies, including one conducted by Maguire et al. where they found that intraparenchymal delivery of human interferon beta by AAV prevented human GBM development and led to tumor regression in xenograft models of disease[17]. In their study, they stress the capacity of transfecting non-tumor cells in the TME as an effective strategy in longer lasting production of transfected viral payload.

Although AAV6 produced CXCL9 clearly improves lymphocyte chemotaxis in vitro, we found that AAV6-CXCL9 monotherapy did not induce robust lymphotaxis into intracranial GBM tumors. The addition of anti-PD-1 ICB dramatically improved lymphocyte trafficking in the combinatorial setting with AAV6-CXCL9, in particular CD8 T cells. These findings speak toward the prospect of T cell sequestration as an auxiliary barrier to trafficking that jointly contributes to lymphopenia in GBM as first reported by Chongsathidkiet and colleagues[7]. Therefore, treatment strategies to boost peripheral lymphocyte counts may be necessary to realize the potential of AAV-based chemotactic therapy. Anti-PD-1 ICB offers one such modality, as peripheral expansion of T cells has been validated as a clinical correlate of response to this immune checkpoint inhibitor in certain cancers[69,70]. GBM is largely refractory to anti-PD-1 ICB including in somatically hypermutated

GBM,[8,71] reciprocally underscoring the need for multimodal treatment strategies. Our findings echo this sentiment, with anti-PD-1 ICB showing minimal improvements in overall survival in the preclinical setting. Previous work done by Pascual-Garcia et al. identified the IL-6 class cytokine Leukemia Inhibitor Factor (LIF) as a potential direct regulator of CXCL9 expression in human GBM preventing recruitment of CD8 + T cells and impairing anti-PD-1 therapy[72]. In their work, there was an inverse relationship between LIF upregulation and CXCL9 expression in human GBM samples, and inhibition of LIF through RNAi or monoclonal antibody therapy could induce upregulation in CXCL9 resulting in improved T cell recruitment. Similar to our data presented here, as a monotherapy, this strategy cannot overcome tumor proliferation. However, when added to anti-PD-1 ICB they observed similar increases in survival and induction of immunologic memory[72]. These data suggest that deficiencies in lymphocyte trafficking and tumor infiltration likely contribute to the problem of immunotherapy resistance, where strategies that upregulate CXCL9 may overcome this barrier. Both strategies have benefits and drawbacks. Peripheral monoclonal antibody administration poses little logistical considerations in terms of safety and feasibility; however, it is unknown if a LIF-targeting monoclonal antibody can sufficiently cross the BBB. As we anticipate heterogeneity among human GBM patients, LIF may not be a constitutively expressed target across all patients. Therefore, inhibition of LIF may not necessarily correlate with upregulation in CXCL9. This is highlighted by Garcia's work, which emphasized that LIF inhibition in the GL261 parent line had no effect but was effective in the GL261N subline with elevated LIF expression levels. Therefore, AAV directed production of CXCL9 at the tumor site, independent of LIF expression, may be applicable to a broader range of GBM patients. Moreover, intratumoral delivery has the capacity to localize the recruitment and migration of T cells to the tumor site as compared to systemic delivery which may redirect or dilute the response of CD8 + T cells.

Another consideration is to combine AAV6-CXCL9 gene therapy alongside adoptive cellular transfer of ex vivo modified T cells, such as CAR T cell therapy. This strategy would bypass host lymphocyte sequestration altogether by direct intravenous delivery of antigen-specific T cells, where tumor-tropic CXCL9 expression would facilitate directed trafficking. Likewise, tumor-specific homing of CAR T remains an unresolved issue for solid tumors[9-11] and may benefit from this particular combinatorial strategy. Both AAV6-CXCL9 and anti-PD-1 ICB also confer secondary mechanisms of action that protect against immune tolerance. CXCL9 is reported to promote lymphocyte differentiation and maturation towards an effector phenotype[73] and anti-PD-1 ICB protects against TME PD-L1 induced lymphocyte exhaustion, thus contributing toward adaptive immune activation and prolongation of cytotoxicity in these cells[74]. Our findings corroborate this, confirming enhanced CD8 T cell cytotoxicity and TCR signaling pathway activation alongside detection of pro-inflammatory chemokines and cytokines associated with adaptive immunity, particularly in the context of combination treatment. Total abrogation of therapeutic efficacy with CD8 lymphocyte depletion validates adaptive immune activation as the principal mechanism of anti-tumor response. Combination virotherapy with pooled AAV vectors targeting different aspects of immunogenicity should also be explored further as a viable multimodal strategy for overcoming GBM immune evasion.

We found that direct intratumor delivery of AAV6 was sufficient to establish tumor-selective production of CXCL9. While intratumor delivery is considered an invasive method of treatment, generating focal expression of chemokine is critical for efficacy as the mechanism by which these secreted molecules work to facilitate immune cell trafficking is by establishing concentration gradients that immune cells expressing the cognate receptor follow, prompting selective infiltration into inflamed tissues[58]. While brain tropic, BBB-crossing serotypes have received significant attention for their ability to transduce CNS

tissue following intravenous delivery[15,16], higher doses are required to maintain sufficient transduction efficiency, and off-target transduction of peripheral tissues remains a consequence of this delivery method[75-78]. Peripheral expression of CXCL9 or other lymphotactic chemokines could counterproductively deter homing to the CNS, reducing the efficacy of this treatment strategy. Furthermore, systemically delivered AAV could also encourage immunological responses resulting in host complement activation and antibody-mediated neutralization, or could prompt adverse toxicity such cytopenias, hepatoxicity, and even neurotoxicity[77-79]. To generate focal AAV transgene expression, virus could be delivered via stereotactic injection into unresectable gliomas, for example in tumors that arise in vital structures of the brain or in the event of recurrence, or into the resection cavity following surgery.

In summary, the use of AAV gene therapy has the potential to disrupt the existing treatment paradigm for GBM which relies on radiation, surgery, and cytotoxic chemotherapy. Systemic administration of immunotherapeutics and single-target chemotherapy agents have shown limited clinical efficacy due to dose-limiting toxicities, the constraints of the blood brain barrier (BBB), and the suppressive nature of the TME. This study combines the excellent safety profile of AAV[13] with focal delivery directly to the TME, bypassing the restrictions and limitations of systemic delivery. AAV biotherapy is minimally invasive, tunable, and enables simultaneous delivery of multiple anti-cancer agents that can be customized to targets unique to each brain tumor. This platform has further application across multiple metastatic tumors where the TME limits the efficacy of immunotherapy.

## Methods
The results presented here comply with all relevant ethical regulations. Ethical approval for the use of human tissue specimens were obtained through the Florida Center for Brain Tumor Research (FCBTR) under the University of Florida Institutional Review Board protocol 201300482. The patient tissues used in this study have an 'Exempt Non-Human Status' approval from the IRB as it is considered non-human in nature, with no identifiers.

All animal experiments complied with local and federal animal welfare standards, and all protocols were independently reviewed and approved by the University of Florida Institutional Animal Care and Use Committee (IACUC).

### Cell culture
KR158B-luc (Kluc) glioma line (provided by Dr. Karlyne M. Reilly, NCI Rare Tumor Initiative, NIH) and GL261 cells have been verified histologically as high-grade glioma, and gene expression analysis confirmed appropriate haplotype background and expression of astrocytoma-associated genes[80]. CT-2A were purchased from Millipore Sigma (cat# SCC194). Primary human glioma cells including L0, L1, L2, CA1, CA2, CA4, CA6, CA7, L23, L26, L31, L34, L38, L47, and HA2 were a kind gift from Dr. Brent A. Reynolds[25]. C8-D1A primary astrocytes were purchased from ATCC (cat# CRL-2541). All cells were cultured in DMEM (Fisher-Scientific) supplemented with 10% FBS (VWR) and 1% Penn-Strep (Life Technologies), and maintained at 37°C in humidified conditions with 5% $CO_2$. At the beginning of the study, cells were expanded, stocks made, and thawed vials were maintained in culture for no more than 3 weeks.

### In vivo studies
Female C57BL/6 J (Strain# 000664), CCR2[RFP]CX3CR1[GFP] (Strain# 032127), GREAT (Strain# 017581), and UBC-GFP (Strain# 004353) mice were purchased from Jackson Laboratory. Animals were maintained at the animal facility of the University of Florida in ventilated cages in a pathogen-free facility in a standard environmentally controlled room, with 50% humidity and 22 C temperature under a 14–10 h light-dark cycle. Standard water and diet were given to the mice. On day 0, 5 x 10⁴

tumor cells suspended in 50% methylcellulose and 50% saline (Fisher-Scientific) were stereotaxically injected into murine brain at a depth of 3 mm, 2 mm lateral to bregma, at a volume of 2 μl in 8–12 week-old animals. On day 5, AAV6 vectors were intratumorally injected in the same coordinates as tumor implantation. For contralateral AAV delivery, AAV6 vectors were injected at a depth of 3 mm, -2mm lateral to bregma. Where indicated, monoclonal antibody treatment (PD-1 ICB, IgG control, CD4 or CD8a depletion) was administered beginning Day 5 via intraperitoneal injection and given every 72 h. Endpoint criteria used in survival analyses are defined by the presentation of neurological symptoms (e.g. decreased grip strength, ataxia, circling, paralysis, and seizure activity), appearance, cranial deformity, and/or decline in body condition score as a result of advanced disease burden, as tumor size is not readily apparent in the orthotopic models utilized in this study. Defined limits were not exceeded. Protocols (201910827, 201907966, 202100000029, and 202300000171) were reviewed and approved by the University of Florida Institutional Animal Care and Use Committee.

### Clinical specimens
De-identified patient tissues were procured by the Florida Center for Brain Tumor Research (FCBTR) under the University of Florida Institutional Review Board protocols 201300482.

### Drug
InVivoMAb anti-mouse PD-1 (cat# BE0146) and InVivoMAb rat IgG2a isotype control (cat# BE0089) monoclonal antibodies were purchased from BioXcell, diluted in Sterile Saline 0.9% solution (Patterson Veterinary Supply, Inc.), and administered via intraperitoneal injection at a dose of 10 mg/kg given every 72 h for a total of 4 doses. InVivoMAb anti-mouse CD8α (cat# BE0061), InVivoMAb anti-mouse CD4 (cat# BE0003-3), InVivoMAb rat IgG2b isotype control (cat# BE0090), InVivoMAb rat IgG2a isotype control (cat# BE0089) monoclonal antibodies were purchased from BioXcell, diluted in Sterile Saline 0.9% solution (Patterson Veterinary Supply, Inc.), and administered via intraperitoneal injection at a dose of 300 μg/mouse given every 72 h for a total of 6 doses.

### AAV protocol
HEK 293 T cells (ATCC cat# CRL3216) were cultured to ~70% confluency in two Cellstacks (Corning cat# 3269) per construct and transfected using PEI 25 k MW (Polysciences cat# 23966-1) for 3 days. The cells were then harvested via shaking and centrifugation until cell pellet was formed. The pellet was then digested with a final concentration of 50 U/mL of Benzonase (Sigma cat# E8263) and 0.5% sodium deoxycholate in a lysis buffer (150 mM NaCl, 50 mM Tris-HCl pH 8.4) for 30 min at 37 °C. Following incubation, the supernatant was supplemented with 5 M NaCl until a 1 M final concentration was achieved. Afterwards, the supernatant was lysed via 3 freeze thaw cycles of -80 °C and 50 °C. The lysate was spun down and supernatant transferred to an ultracentrifuge tube (Beckman cat# 342414), where it is layered with discontinuous layers of iodixanol (Accurate Chemical cat# AN1114542) to separate out viral particles from the supernatant. This was spun for 1 h at 18 °C at 350,000xg. The viral particles were isolated and removed, then washed four times in a dialysis column (Millipore cat# UFC910024) with PBS before being finally purified in a sterile filtration column (Millipore cat# UFC30DV00).

### AAV quantification
The viruses were titrated by quantitative PCR (Bio-Rad CFX384) using custom probes designed to target the ITR sequences. First, 1 uL of the virus was treated with DNAseI (Thermo Fisher cat# 18068015) for 15 min at room temperature, inactivated by heat and EDTA, protein coat of virus digested with Proteinase K (Thermo Fisher cat# 25530049) and finished with a second heat-inactivation step.

Following incubations, the sample was diluted and mixed with a Taqman PCR Master Mix (Thermo Fisher cat# 4352042) and the custom designed probes (Thermo Fisher cat# 4332078). The probe sequences were as follows: Forward –GGAACCCCTAGTGATGGAGTT, Reverse –CGGCCTCAGTGAGCGA, Probe –CACTCCCTCTCTGCGCGCTCG. The samples were then compared to a standard curve consisting of a linearized plasmid with ITRs from a range of 1e4–1e8 genomic copies per mL. The samples were then run on a standard program of 10 min denature at 95°C, then cycled 39 times at 95 °C at 1min and 60 °C at 30 s.

### Lentiviral transduction of tumor cells
RFP labeled GL261 and KR158 tumor cells were transduced with a LentiBrite RFP Control Lentiviral Biosensor (Millipore-Sigma, cat# 17-10409), MOI 50. Following cell expansion, RFP-positive cells were FACS sorted using a BD FACSAria-II cell sorter, yielding RFP-stable tumor cells. GFP labeled GL261 tumor cells were produced in the same manner using LentiBrite GFP Control Biosensor (Millipore-Sigma, cat # 17-10387).

### Proteome arrays
Following resection, the right hemisphere (cerebellum removed) of murine brain (tumor-containing) were transferred to 1.5 mL microtubes, snap frozen in LN2, and stored at -80°C until lysis. De-identified flash frozen patient GBM tissue was procured from the FCBTR. Tissue shavings were collected on dry ice and transferred to 1.5 mL microtubes. 300–500 μl PBS containing 1 x Halt™ Protease/Phosphatase inhibitor (Thermo Fisher) and 1% Triton X-100 (Sigma) was added to samples and transferred to wet ice. Tissue was lysed manually using a 20-gauge needle attached to a 1 mL syringe followed by vortexing every 5 min for 30–60 min. Supernatant was collected following centrifugation at 10,000 x g at 4 °C, and assayed for protein concentration using a NanoDrop Spectrophotometer. 0.75 mg of each human sample was used for the Human Chemokine Array Profiler (R&D Systems, ARY017), and 1 mg of each murine sample was used for the Mouse XL Cytokine Array (R&D Systems, ARY028) following manufacturer's instruction. Images were captured using BioRad ChemiDoc MP Imaging System with ImageLab 6.1 software over a series of exposure times. Mean voxel intensity per capture antibody was calculated using Imaris x64 v9.7.0, and protein signal was normalized against internal reference controls. Detected protein and predicted receptor interactions were analyzed and visualized using Circos®[65].

### ELISA
Tissue specimens were collected at 1 and 2 weeks post-AAV6 injection. Peripheral blood was taken from the anterior vena cava, centrifuged at 10,000 x g x 10 min @ RT, and serum collected and stored at -80C. Whole brain was resected, cerebellum removed, and divided into the tumor-bearing (AAV6 injected) and contralateral hemispheres. Naive brain and serum were collected and used to set the baseline for both week 1 and week 2 datasets. Tissue was snap frozen and stored at -80 °C until lysis. Tissue was lysed using RIPA buffer containing 2x Halt protease/phosphatase inhibitor cocktail (Thermo Fisher) with manual dissociation performed using a 20-gauge needle attached to a 1 mL syringe followed by vortexing every 5 min for 30–60 min, and maintained on ice. Following lysis, tissue samples were centrifuged at 12,000xg @ 4 °C x 10 min. Supernatant was collected, and assayed for protein concentration using a NanoDrop. Protein concentrations were adjusted using RIPA buffer. MIG/CXCL9 ELISA (Thermo Fisher) performed according to manufacturer protocol. Serum diluted 1:2 using Assay Diluent B. Tissue sample concentration: 2 mg. All samples run in duplicate. ELISA detection of CXCL9 in competitive co-culture assays were performed on undiluted media collected from respective cell chambers prior to co-culture.

## Cell proliferation assay

Cells were seeded at $2 \times 10^4$ cells per well in 24 well plates in complete media (day 0). On days 1, 3, and 5 cells were detached from the plate using 0.25% trypsin (Gibco), and counted using a ViCELL Analyzer (Beckman Coulter).

## 3D tissue clearing and immunolabeling

Brain tissue was collected after cardiac perfusion with cold saline followed by PBS supplemented with 4% acrylamide (Sigma-Aldrich), 0.05% N,N′-methylenebisacrylamide (Sigma-Aldrich), 4% paraformaldehyde and 0.25% VA-044 (TCI America). Tissues were stored at 4 °C for 3 days to allow hydrogel permeation of tissues. Following hydrogel polymerization at 37ºC x 3 h, whole brain was sectioned to 2 mm and passively cleared over 3–7 days with PBS containing 200 mM boric acid (Sigma-Aldrich) and 4% sodium-dodecyl-sulfate (Fisher-Scientific), pH 8.5 at 50ºC. After clearing, samples were washed in PBS with 0.1% Triton X-100 for 2 days, and immunostained at 4 °C for 2 days with the following antibodies and stains: GFAP (Thermo Fisher, cat# PA1-10004, 1:50 dilution), CD45 (Thermo Fisher, cat# 14-0451-82, 1:50 dilution), anti-chicken Alexa Fluor™ 647 antibody (Thermo Fisher, cat# A-21449, 1:200 dilution), anti-rat Alexa Fluor™ 568 (Thermo Fisher, cat# A-11077, 1:200 dilution), and either DAPI (Sigma-Aldrich) or DRAQ5 (Thermo Fisher) nuclear dye. Samples were whole-mounted onto slides using 62% 2,2′-Thiodiethanol (Sigma-Aldrich). Images were acquired using a Nikon A1RMP confocal microscope and analyzed using Imaris x64 v9.7.0 software. Detection of lymphocyte infiltration was performed as follows: CD3+ lymphocytes were isolated from naïve donor UBC-GFP spleens using the MojoSort™ mouse CD3 T cell Isolation Kit (Biolegend, cat# 480024). These were adoptively transferred ($1 \times 10^6$/mouse) into sham control or combination AAV6-CXCL9 plus anti-PD-1 ICB treated mice harboring GL261 tumors on study day 11 via intravenous tail vein injection. Brain tissue was resected on study day 15, and prepared using the method described above. Whole brain 3D tissue clearing, immunolabeling, and light-sheet imaging was performed by LifeCanvas Technologies.

## Tissue dissociation and flow cytometry

Brain tissue was digested using the Multi-tissue Dissociation Kit (Miltenyi Biotec) on a gentleMACS™ Octo Dissociator with heat, followed by sample clean-up using Debris Removal Solution (Miltenyi Biotec) according to manufacturer's protocol. Tumor-infiltrating leukocytes were isolated using CD45 microbeads (Miltenyi Biotec) filtered through LS columns (Miltenyi Biotec) on a QuadroMACS Separator (Miltenyi Biotec) according to manufacturer's protocol. Blood samples were collected from the anterior vena cava, and RBC lysis performed using Pharm Lyse solution (BD Biosciences) per manufacturer's protocol. Samples were washed 2x with cold PBS. Unstained cells were reserved for unlabeled and FC controls, and dead cells were labeled with Zombie NIR™ Fixable Viability Kit (Biolegend) according to manufacturer's protocol. Cells were washed 2x in PBS containing 0.5% BSA (Sigma) and 2 mM EDTA (Thermo Fisher) FC buffer and blocked for 10 min on ice using TruStain FcX (Biolegend) prior to cell surface antigen labeling with the following antibodies: CD45-APC (Biolegend, cat# 103112, 0.20 μg per 10^6 cells), CD3-FITC (Biolegend, cat# 100204, 1.0 μg per 10^6 cells), CD4-PE (Biolegend, cat# 100408, 0.20 μg per 10^6 cells), CD8-BV421 (Biolegend, cat# 100738, 0.50 μg per 10^6 cells) for 45 min on ice. For astrocyte detection, cells were fixed and permeabilized using True-Nuclear™ Transcription Factor Buffer Set (Biolegend) following manufacturer's protocol following debris removal step, with no CD45 microbead isolation. Samples were labeled with Zombie NIR™ Fixable Viability Kit (Biolegend), blocked with TruStain FcX (Biolegend), and immunolabeled with GFAP-APC (Thermo Fisher, cat# 51-9792-82, 0.50 μg per 10^6 cells). Following immunolabeling, all samples were washed 2x with FC buffer and analyzed using a BD FACSymphony A3 flow cytometer.

## In vitro chemotaxis

GL261 or C8-D1A cells were plated in 24-well dishes at $1 \times 10^5$/well in pre-warmed complete media. AAV6-EGFP or AAV6-CXCL9 (RFP +) was added at a final concentration of $10^5$ VGS. Twenty-four hours following transduction, cells were transferred into the outer chambers of μ-Dishes with 3-well culture inserts (Ibidi), $10^4$, suspended in 15 μl of growth-factor reduced Matrigel® (Corning). 40 μl of complete media was added following polymerization for 10 min at 37ºC in humidified conditions with 5% $CO_2$. CD3 + T cells were isolated from naïve C57BL/6 mouse spleen (8–12 weeks) using MojoSort CD3 T cell isolation kit (Biolegend) per manufacturer's protocol. T cells were labeled with Cell Trace Violet dye (CTV) (Thermo Fisher) per manufacturer protocol. $1 \times 10^4$ labeled T cells were suspended in 15 μl cold growth-factor reduced Matrigel® (Corning), and added to the μ-Dish center well. Following polymerization as described above, media was removed from all wells, and 3-well insert was carefully removed. The gap between wells was filled with additional Matrigel to form a continuous substrate, and allowed to polymerize for 20 min. Complete media was added to cover cells, and incubated at 37ºC in humidified conditions with 5% $CO_2$. IF images were acquired using a Nikon A1RMP confocal microscope at indicated time points (1- and 24 h) following co-culture, and T cell chemotaxis was quantified as the number of migratory cells (CTV +) visible in either the EGFP or CXCL9 (RFP +) transduced tumor/astrocyte field.

## Single cell RNA sequencing, quality control, NanoString and data analysis

Following whole brain resection, cerebellar tissue was removed and the right hemisphere collected for processing. Tissue dissociation and CD45 TIL isolation was performed as described under *Tissue Dissociation* above. The cells directly after isolation were washed with PBS and 0.04% bovine serum albumin two times and filtered with 40-μm cell strainer. Cells were collected by centrifugation at 500xg for 5 min and subsequently counted with hemocytometer. Cells were diluted in ice-cold PBS containing 0.04% BSA at a density of 1000 cells/μL. The final cell suspension volume equivalent to 8000 target cells was used for further processing. Cells were loaded into a Chromium NextGEM Chip G (10x Genomics, Pleasanton, California) and processed in Chromium X following the manufacturer's instructions. Preparation of gel beads in emulsion and libraries were performed with Chromium Next GEM Single Cell 3′ Kit v.3.1 (Dual Index) according to User Guide provided by the manufacturer. Libraries quality and quantity were verified with 2200 TapeStation (Agilent technologies, USA). Libraries were pooled based on their molar concentrations. Pooled library was sequencing on the NovaSeq 6000 instrument (Illumina, San Diego, California). For sequencing 3′ gene expression libraries we used following read length: Read 1–28 cycles; i7Index-10 cycles; i5Index-10 cycles; Read 2–90 cycles. Raw base call (BCL) files generated by NovaSeq 6000 sequencer were processed using Cell Ranger software (10X Genomics, version 7) for demultiplexing, barcode processing, and single-cell 3′-gene counting. Mouse genome reference GRCm38 was used for sequence alignment using STAR aligner. A read was considered exonic, if at least 50% of it mapped to an exon, intronic (if it was non-exonic and intersected an intron), or intergenic otherwise. For reads that aligned to a single exonic locus but also aligned to 1 or more non-exonic loci, the exonic locus was prioritized and the read was considered to be confidently mapped to the exonic locus. Cell Ranger also aligned exonic reads to annotated transcripts. An annotated transcript that aligned to the same strand was considered to be confidently mapped to the transcriptome. These confidently mapped reads were used for unique molecular identifier (UMI) counting and subsequent analysis to generate h5 files. The h5 file of each sample was then processed with Partek Flow analysis software (version 10). Cells meeting the following quality control (QC) parameters were included in the analysis: total reads between 1000–33,649; expressed genes between 187–5464; mitochondrial reads percentage < 20%. Following this selection, we obtained 48159 cells that passed QC

filters. Next, features were filtered in order to include only genes expressed in >0.01% of cells and 20,785 genes were retained. UMI counts were then normalized following Partek® Flow® recommendations: for each UMI in each sample the number of raw reads was divided by the number of total mapped reads in that sample and multiplied by 1,000,000, obtaining a count per million value (CPM), the normalized expression value was log-transformed. Starting from the normalized data node, we performed clustering analysis for each sample separately by means of graph-based clustering task in Partek® Flow® software which employs the Louvain algorithm. Clustering analysis was done based on the first 100 principal components. To visualize single cells in a two-dimensional space, we applied Uniform Manifold Approximation and Projection (UMAP) plot using the first 50 principal components for each sample separately and for the entire data set. Cell types were determined by the expression of marker genes that define specific cell types (Supplemental table 1). Pathway enrichment analysis for tumor cells and immune cells was performed with AUCell algorithm using the NanoString nCounter Immune Exhaustion panel. Interactions between immune populations were analyzed and visualized using the CellChat algorithm[38]. The pheatmap package was used for unsupervised hierarchical clustering to create heatmaps[81]. Clustering of T cell subpopulations was performed by taking predefined "CD8 T cells," "CD4 T cells," and "Treg cells" and assembling them into a Seurat object. Principal Component Analysis (PCA) and the UMAP algorithm were then employed. Cell types were determined based on previously defined differentially expressed genes within each cluster[82]. The Seurat feature plot function was utilized to examine the expression of the Cxcr3 gene. The scRNAseq data files generated in this study have been deposited in the open-access Genome Sequence Archive database under the accession code PRJCA022912.

### Statistical analysis
Statistical analyses performed using GraphPad Prism 9 as described in figure legends. Significance determined as p < 0.05. Voxel-based co-localization was established using Imaris x64 v9.7.0 using the Coloc module with automatic threshold selection. For survival studies, animals were randomized prior to treatment.

### Reporting summary
Further information on research design is available in the Nature Portfolio Reporting Summary linked to this article.

## Data availability
The single cell RNA sequencing data generated in this study have been deposited in the open-access Genome Sequence Archive database under the accession code PRJCA022912. The remaining data are available within the Article, Supplementary Information or Source Data file. Source data are provided with this paper.

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

## Acknowledgements

This work is supported in part by the Wells Brain Tumor Research Fund (DAM), the Adam Michael Rosen Research Fund (DAM), the McKnight Brain Institute Fellowship Award (CAV), and the Circle of Hope Foundation (CAV). Research reported in this publication was also supported by the National Center for Advancing Translational Sciences of the National Institutes of Health under University of Florida and Florida State University Clinical and Translational Science Awards (TL1TR001428 and UL1TR001427), and the University of Florida Interdisciplinary Center for Biotechnology Research with funding provided by NIH Grant (1S10OD020026). Research reported in this publication was also supported by the UF Health Cancer Center, supported in part by state appropriations provided in Fla. Stat. § 381.915 and the National Cancer Institute of the National Institutes of Health under Award Number P30CA247796. The content is solely the responsibility of the authors and does not necessarily represent the official views of the National Institutes of Health or the State of Florida. Figures 1g, 2c, 3d, and 8 were created using BioRender.com, released under a Creative Commons Attribution-NonCommercial-NoDerivs 4.0 International license.

## Author contributions

C.A.V., D.A.M., T.E.G.: conceptualization, design, project supervision, administration, funding acquisition. C.A.V., J.A.P., S.L.C., O.Y., and C.Y.: methodology, data curation, formal analysis. A.B., B.P.D., R.R., V.S.T., K.K., D.H.R., A.G., H.S.F., Y.R., L.B.M.H., and F.L.W.: data curation. C.A.V., B.P.D., and D.A.M.: writing, review, editing. All authors contributed to critical reading and revision of the manuscript. All authors have read and agreed to the published version of the manuscript.

## Competing interests

The authors declare no competing interests.
