## [Peer Review File · Nature Communications]

Adeno-associated virus delivered CXCL9 sensitizes glioblastoma to anti-PD-1 immune checkpoint blockadeREVIEWER COMMENTS

Reviewer #1 (Remarks to the Author): with expertise in astrocytes, glioblastoma

The manuscript by von Roemeling et al. aims to develop an AAV (adeno-associated virus)-based virotherapy for treating glioblastoma (GBM). By using AAV to overexpress CXCL9 in tumor-associated astrocytes, there was increased infiltration of CD8-positive lymphocytes in mouse GL261 and KR158 xenograft glioma. The treatment sensitized mouse GBM to anti-PD-1 immune checkpoint blockade (ICB). The combination of the AAV-CXCL9 and anti-PD-1 ICB treatment induces a wide range of change of the immunological landscape of the GBM tumor microenvironment (TME). Overall, I think this study demonstrates the positive therapeutic effect of the proposed strategy and discovers the immunological signatures related to the effect. However, based on the concerns raised by the author about the underlying mechanism, long-term immune memory, and the safety of this potential virotherapy, I think there are some issues needed to be addressed to resolve the possible questions and concerns.

A critical issue that affects the significance of the manuscript is that the authors did not refer the article “LIF regulates CXCL9 in tumor-associated macrophages and prevents CD8+ T cell tumor-infiltration impairing anti-PD1 therapy” published on Nature Communications in 2019 (DOI: 10.1038/s41467-019-10369-9). This article demonstrated several critical findings echoing the results in the manuscript. This article showed the critical role of CXCL9 in the recruitment of CD8+ T cell to GBM TME. By blocking LIF using monoclonal antibody given systemically, CXCL9 expression was increased in tumor-associated macrophages and sensitized mouse GBM to anti-PD-1 ICB. The authors should refer the finding in this article and compare the similarity and the difference between the previous study and their own result. The authors should also discuss the pros & cons of these two therapeutic strategies focusing on the expression of CXCL9.

Another important issue is that the authors claim that “the tumor-associated astrocytes may be selectively susceptible to AAV6 transduction” (Line 362 – 363). However, this argument is contradicted to the result shown in Figure 1e, f. In this experiment, the transduction ratios of AAV6-EGFP were the same in the naïve astrocytes as in the tumor-associated astrocytes. Whether AAV6 transduction is limited to the tumor-associated astrocytes is an important factor that affects the efficiency and the safety of the

virotherapy, the author should be very cautious when they are interpreting the result. In addition, the finding that transgene expression was limited to the immediate tumor area (Line 361 - 362) might simply be because the injection site of the AAV is the same as the injection site of tumor cells. To rule out this possibility and claim that tumor-associated astrocytes are selectively susceptible to AAV6 transduction, extra experiment will be necessary to prove the claim.

There are some minor issues needed to be addressed or fixed to improve the data interpretation and presentation:

1. The images in the lower panels of Figure 1e do not look like matching the area marked by yellow dash line in the upper panel of Figure 1e. The area outlined by the yellow dash line in the upper panel of Figure 1e may need to move to bottom-right location relative to the current location.
2. In the figure legend of Figure 1d, it mentioned "RFP+ tumor cells in light blue pseudocolor". Please check the figure to confirm that whether it is in light blue or in white pseudocolor.
3. The digits of p-value in extended figure 1f seems to be cut off during editing.
4. A typo in line 673, "40- μ n cell drainer" should be "40- μ m cell drainer".

Reviewer #2 (Remarks to the Author): with expertise in glioblastoma, cancer immunology

In the study by von Roemeling et al., the authors exploit AAV to deliver CXCL9 into the immunosuppressive experimental glioblastoma microenvironment of two syngeneic glioblastoma models. Convincingly, the authors selected AAV6 following screening of 29 unique AAV serotypes. AAV6 has been known previously to target astrocytes. Although the minority of CXCL9 expressing cells following AAV treatment were tumor cells in vivo and the expression in the latter was overall of transient nature, the authors observed increased CD8+ T cell abundance in tumors following the combinatorial treatment of AAV-CXCL9 and anti-PD1. Robust preclinical survival data are presented for two experimental models. In re-challenge experiments, AAV-CXCL9 and anti-PD1-treated animals were protected. Response was CD8-T cell dependent. The exploratory scRNA-seq analysis is highly descriptive by suggesting an increased cellular crosstalk of lymphocytes to other cells of the TME. None of the suggested pathways has been experimentally proven to be predictive for response to

AAV-CXCL9 and anti-PD1 in the exploited models.

- (1) The concept of local delivery (to tumor-associated astrocytes) needs to be experimentally proven. I would expect that there is some tumor control even if the tumor is growing in the contralateral hemisphere and AAV is delivered to non-tumoral astrocytes
- (2) Figure 3c. There was no competitive co-culture comparing the migratory capacity of T cells towards CXCL9-expressing GL261 and astrocytes depicted.
- (3) Figure 3e-h. Anti-PD1 should unleash peripheral and intratumoral T cells and thus T cell proliferation. The authors measure T cell abundance in tumors but not T cell infiltration. This should be carefully taken into consideration.
- (4) Treating mice at day 5 is very early, at this time point, in e.g., preclinical MRI, tumors are macroscopically not visible. How is intratumoral injection confirmed then? The authors should repeat their therapeutic treatments using AAV6-CXCL9 at day 10-12.
- (5) Fig 4c/d: Where are the biological controls (AAV6-GFP?) for the IF?
- (6) What is the role of CD4+ T cells when applying AAV-CXCL9?
- (7) It would be important to assess, how CXCL9-expressing astrocytes interact with T cells

Reviewer #3 (Remarks to the Author): with expertise in AAV-mediated gene therapy of the brain

CXCL9 recombinant adeno-associated virus (AAV) virotherapy sensitizes glioblastoma (GBM) to anti-PD-1 immune checkpoint blockade by Roemeling et al. seeks to develop a novel AAV-based therapy using tumor stromal cell expression of the chemokine CXCL9 in combination with anti-PD-1 antibodies for glioblastoma (GBM) therapy.

The authors tested several capsids for transduction of glioma cells in culture and AAV6 was one of the best performers. Intratumoral injection of AAV6-GFP into mice bearing GBM tumors, yielded some transduction of tumor cells, but the vast majority of GFP positive cells for astrocytes around/within the tumor. The authors then tested the use of AAV-CXCL9 as a therapeutic agent for GBM. In vitro assays demonstrated chemotaxis of T cells towards AAV-CXCL9 transduced tumor or astrocytes. In vivo, low levels of CXCL9 could be detected in the tumor/AAV-CXCL9 injected side of the brain, but not the contralateral side (suggesting a relatively low level of transduction). In two glioma models, the authors also found that

AAV6-CXCL9 combined with anti-PD-1 antibody increase CD8 T cell recruitment to the tumor.

Next the authors performed a therapeutic study with AAV6-CXCL9 combined with anti-PD-1. A modest but significant increase in median survival was observed in two glioma models with the combined treatment over controls, with long-term survivors (50% and 25%) observed in both models. They also found that CD8 T cells were crucial to the survival using a CD8 depletion experiment. The authors next performed extensive RNAseq analysis to observe differences in specific cell type signatures between the different control groups and made some interesting observations.

The in vitro and in vivo AAV cancer gene therapy experiments have been done very well, and the reviewer is appreciative of the amount of labor that went into this work. Also, the strategy of using AAV vectored-chemotaxis to recruit T cells into the tumor environment along with anti-PD-1, seems like a feasible strategy, and the authors provide proof of concept data of efficacy.

This notwithstanding, a major novelty of the paper appears to be the observation that “AAV targeting of tumor-associated cells may carry several distinct advantages over directly targeting cancer cells....” This statement appears to suggest that the authors were discovering/proposing something novel, while it is not (see comment 1 below).

Unfortunately, the existing literature (which includes AAV transduction of astrocytes for GBM therapy) greatly reduces the novelty of the work for this journal’s broad audience. It does appear, however, that the combined use of AAV-based chemokine expression to bring T cells into the tumor adds some novelty to this study.

Comments:

Comment 1: This strategy of using the tumor stromal microenvironment for expression of AAV-vectored transgenes has been used by other groups (see below) successfully in past published work, however the authors did not appear to be aware of this. Proper citations of the prior work should be carefully assessed by the authors.

Examples:

<https://pubmed.ncbi.nlm.nih.gov/36217021/>

<https://pubmed.ncbi.nlm.nih.gov/37172581/>

<https://www.ncbi.nlm.nih.gov/pmc/articles/PMC2863297/>

<https://www.ncbi.nlm.nih.gov/pmc/articles/PMC6454875/>

<https://pubmed.ncbi.nlm.nih.gov/25501993/>

Comment 2: It seems that the AAV6 transduction of tumor-associated astrocytes was an unexpected observation by the authors and that expression levels of CXCL9 by this system were quite low. To improve the outcome of the combination therapy, it is likely that the system needs further optimization at the capsid and transgene expression cassette levels.

Reviewer #4 (Remarks to the Author): with expertise in glioblastoma, cancer immunology

This is an interesting and timely study from a well-established brain tumor group at the University of Florida. It is well-known that malignant gliomas have greatly reduced numbers of tumor infiltrating lymphocytes (TIL), but yet these cells are necessary for functional anti-tumor immunity and immunotherapy. Van Roemeling, et.al, try to address this significant gap through the AAV recombinant production of CXCL9, a potent chemokine known to recruit T cells into the tumor microenvironment (TME). While CXCL9 by itself, does not seem to be able to elicit significant migration of T cells into the TME, in conjunction with PD-1 mAb blockade, these investigators do see elevated CD8+ T cells in the tumor, and was associated with extended survival in mice bearing two different models of malignant gliomas. The efficacy of the combination treatment was completely dependent on CD8 cells. Using several high dimensional technologies (e.g. single cell RNAseq), this group also showed how the combination of AAV-encoded CXCL9 + PD-1 mAb resulted in enhanced T cell activation, upregulation of immune cell exhaustion and cellular crosstalk with other tumor infiltration immune cells. Such pre-clinical studies form the basis of a better understanding of how TIL density is limiting in this kind of tumor, and outline a clinically translatable strategy to target this gap. While significant and important for the field, a few issues ought to be clarified to best articulate the overall message and interpretation of the data. These issues are outlined below:

Major Comments.

1. Fig 2. It is interesting that this AAV preferentially infects astrocytes in vivo. Is that really the cell type that they would want to move forward with and use? I would think that cells of

the myelo-monocytic lineage would be more effective to target, as they are already known to secrete these kinds of chemokines during productive anti-tumor immune responses. Did the authors ever identify different AAV's that could infect myeloid cells?

2. Fig 3. It would be useful to understand better how the in vitro migration assay was set up? This is not articulated well in the text or the legend.

3. Fig. 3. The authors nicely show an increased CD8+ TIL infiltration into two distinct pre-clinical models of glioma with AAV-CXCL9 + PD-1 mAb. It would be important, though, to show the receptor expression on these CD8 T cells to argue that the CXCL9 was acting through its cognate receptor. Do particular T cell subsets express CXCR3?

4. Fig 6. It is difficult to understand what question and/or new information the Nanostring adds to this story? It is also difficult to visually understand the significant changes in Fig. 6f. Maybe better just to highlight or focus on the selected cell type gene expression changes? In addition, it is not clear how these studies were done? I don't see any methods for the Nanostring analyses and whether they used single cell suspensions or bulk tissue?

5. Fig. 7. Same basic methodology question here. How were these studies done? It is not clear from the text.

REVIEWER COMMENTS

Manuscript Number: NCOMMS-23-53925-T

Manuscript Title: CXCL9 recombinant adeno-associated virus (AAV) virotherapy sensitizes glioblastoma (GBM) to anti-PD-1 immune checkpoint blockade

Dear reviewers,

We sincerely thank you for your time and effort put into reviewing our manuscript submission, “*CXCL9 recombinant adeno-associated virus (AAV) virotherapy sensitizes glioblastoma (GBM) to anti-PD-1 immune checkpoint blockade*”. We found your insights invaluable, and have made a significant effort to address the suggestions and concerns brought forth by each of you. Given the large volume of added studies, we did re-organize several of the figures to improve the narrative flow. Please find our detailed responses to each of your comments below in blue text, which includes, where applicable, a summary of newly added experiments and/or changes made to the manuscript text in bold font. If any of our responses are unclear, or you wish additional changes, please let us know.

Reviewer #1 (Remarks to the Author): with expertise in astrocytes, glioblastoma

The manuscript by von Roemeling et al. aims to develop an AAV (adeno-associated virus)-based virotherapy for treating glioblastoma (GBM). By using AAV to overexpress CXCL9 in tumor-associated astrocytes, there was increased infiltration of CD8-positive lymphocytes in mouse GL261 and KR158 xenograft glioma. The treatment sensitized mouse GBM to anti-PD-1 immune checkpoint blockade (ICB). The combination of the AAV-CXCL9 and anti-PD-1 ICB treatment induces a wide range of change of the immunological landscape of the GBM tumor microenvironment (TME). Overall, I think this study demonstrates the positive therapeutic effect of the proposed strategy and discovers the immunological signatures related to the effect. However, based on the concerns raised by the author about the underlying mechanism, long-term immune memory, and the safety of this potential virotherapy, I think there are some issues needed to be addressed to resolve the possible questions and concerns.

A critical issue that affects the significance of the manuscript is that the authors did not refer the article “LIF regulates CXCL9 in tumor-associated macrophages and prevents CD8+ T cell tumor-infiltration impairing anti-PD1 therapy” published on Nature Communications in 2019 (DOI: 10.1038/s41467-019-10369-9). This article demonstrated several critical findings echoing the results in the manuscript. This article showed the critical role of CXCL9 in the recruitment of CD8+ T cell to GBM TME. By blocking LIF using monoclonal antibody given systemically, CXCL9 expression was increased in tumor-associated macrophages and sensitized mouse GBM to anti-PD-1 ICB. The authors should refer the finding in this article and compare the similarity and the difference between the previous study and their own result. The authors should also discuss the pros & cons of these two therapeutic strategies focusing on the expression of CXCL9.

We thank the reviewer for this suggestion and information. We agree with the significance of this previous work and have incorporated it into the discussion. As the primary takeaway from Pascual-Garcia and team’ manuscript, targeted inhibition of LIF to restore CXCL9 expression and subsequent sensitization to anti-PD-1 inhibitions affirms that tumor expression of CXCL9 is a viable therapeutic strategy to improve anti-cancer activity. We similarly show that anti-tumor activity is perpetuated by tumor-infiltrating CD8 lymphocytes. Where our two studies differ significantly is the method for producing tumor-specific expression of CXCL9, with Pascual-Garcia and team’ using a monoclonal antibody targeting of LIF to

restore CXCL9 expression by tumor-associated macrophages and our group leveraging AAV to delivery a CXCL9 transgene to the tumor microenvironment. We acknowledge that there are some strengths and advantages in comparing these two respective studies. While intravenous systemic delivery of a therapeutic antibody is a considerably more straightforward delivery method, there are some tactical challenges in terms of blood-brain-barrier penetration and achieving distributed therapeutic dosing within brain tumors. Intratumor AAV delivery carries the advantage here. Moreover, Pascual-Garcia and team showed that tumors with little or no LIF expression responded sub-optimally to treatment (parent GL261 model) as compared to high expressing tumors (GL261N model), indicating that tumor heterogeneity may be a limitation to LIF inhibition alone. Direct upregulation of CXCL9 using AAV vectors could overcome this limitation, with broader applicability across genetically diverse GBM tumors.

The manuscript amendment to address this concern can be found on **Page 9, Lines 463-481**.

Another important issue is that the authors claim that “the tumor-associated astrocytes may be selectively susceptible to AAV6 transduction” (Line 362 – 363). However, this argument is contradicted to the result shown in Figure 1e, f. In this experiment, the transduction ratios of AAV6-EGFP were the same in the naïve astrocytes as in the tumor-associated astrocytes. Whether AAV6 transduction is limited to the tumor-associated astrocytes is an important factor that affects the efficiency and the safety of the virotherapy, the author should be very cautious when they are interpreting the result.

We agree that the statement of enhanced sensitivity in only tumor-associated astrocytes would be an over simplification of the processes without further direct investigation of these cells vs non-reactive astrocytes. We have run into challenges testing this experimentally with *in vivo* delivery of AAV6, which requires a stereotactic injection that prompts a localized inflammatory event, leading to the formation of a reactive phenotype in astrocytes in naïve animals. The speculation here is that the brain is responding to this insult through astrocytic activation akin to the glial scarring seen with parenchymal injury. In this sense these cells may be closer to reactive astrocytes themselves which could explain the shared transduction efficiency. What we have experimentally confirmed is that AAV6 transgene signal remains tumor tropic as evidenced in Figure 2a-b. As part of our response to one of the other reviewers, we have also added whole brain imaging of AAV6 transduction in larger tumors that reinforce the focality of transgene signal following intratumor delivery (extended data Figure 5h), which we find encouraging from a safety perspective. To clarify this point we have amended the description of claiming selective ‘tumor-associated’ astrocytes to simply astrocytes so as not to overstate the specificity of this event only being seen in tumor-associated astrocytes.

The modified text can now be found beginning on **Page 8, Lines 427-435**, and reads as follows: “Despite this discrepancy, we found that AAV6-transduced astrocytes confer a high degree of tumor tropism, where transgene expression was limited to the immediate tumor area. Whether this is due to localization of AAV delivery at the injection site, which is an advantage of direct inoculation over systemic administration, or through changes to the reactive astrocytes surrounding the tumor that lead to increased transfection susceptibility is an area of active investigation. Our study additionally shows that AAV6 inoculation *in vivo* in non-tumor bearing mice preferentially transfects astrocytes, potentially indicating the inoculation itself is an injury/inflammatory event that sequesters the therapy at the site of delivery. Altogether our findings suggest that astrocytes may be selectively susceptible to AAV6 transduction as compared to the other cells constituting the tumor microenvironment that reside within the treatment field of the viral administration.”

In addition, the finding that transgene expression was limited to the immediate tumor area (Line 361 - 362) might simply because the injection site of the AAV is the same as the injection site of tumor cells. To rule out this possibility and claim that tumor-associated astrocytes are selectively susceptible to AAV6 transduction, extra experiment will be necessary to prove the claim.

We agree that it is premature to claim that localization occurs solely through enhanced selective transfection of AAV6 in reactive astrocytes. What we have observed *in vivo* is a significant increase in the transfection of astrocytes vs the other cellular representatives of the TME. This comparison of the cells within the “treatment field” of viral administration is highly suggestive of favorable transfection of astrocytes compared with tumor, immune infiltrates, or microglia. The localized effect of this administration is one of the potential strengths of the direct inoculation strategy as it limits the potential for systemic or off target activation. We have added the text to describe this relationship more clearly as outlined in our response to the previous query (**Page 8, Lines 427-435**).

There are some minor issues needed to be addressed or fixed to improve the data interpretation and presentation:

1. The images in the lower panels of Figure 1e do not look like matching the area marked by yellow dash line in the upper panel of Figure 1e. The area outlined by the yellow dash line in the upper panel of Figure 1e may need to move to bottom-right location relative to the current location.

Thank you for identifying this issue- we suspect this may have occurred with the migration of the files from powerpoint to word. We have corrected this.

2. In the figure legend of Figure 1d, it mentioned “RFP+ tumor cells in light blue pseudocolor”. Please check the figure to confirm that whether it is in light blue or in white pseudocolor.

We reviewed the software description of the colors utilized for pseudocolor representation and have changed the legend to reflect this based on their description this is now described as “gray pseudocolor”.

3. The digits of p-value in extended figure 1f seems to be cut off during editing.

Thank you for identifying this data migration issue, it has been corrected in the current version.

4. A typo in line 673, “40- μ n cell drainer” should be “40- μ m cell drainer”.

We have made this correction, which now can be found on **Page 37, Line 1019**.

Reviewer #2 (Remarks to the Author): with expertise in glioblastoma, cancer immunology

In the study by von Roemeling et al., the authors exploit AAV to deliver CXCL9 into the immunosuppressive experimental glioblastoma microenvironment of two syngeneic glioblastoma models. Convincingly, the authors selected AAV6 following screening of 29 unique AAV serotypes. AAV6 has been known previously to target astrocytes. Although the minority of CXCL9 expressing cells following AAV treatment were tumor cells in vivo and the expression in the latter was overall of transient nature, the authors observed increased CD8+ T cell abundance in tumors following the combinatorial treatment of AAV-CXCL9 and anti-PD1. Robust preclinical survival data are presented for two experimental models. In re-challenge experiments, AAV-CXCL9 and anti-PD1-treated animals were protected. Response was CD8-T cell dependent. The exploratory scRNA-seq analysis is highly descriptive by suggesting an increased cellular crosstalk of lymphocytes to other cells of the TME. None of the suggested pathways has been experimentally proven to be predictive for response to AAV-CXCL9 and anti-PD1 in the exploited models.

(1) The concept of local delivery (to tumor-associated astrocytes) needs to be experimentally proven. I would expect that there is some tumor control even if the tumor is growing in the contralateral hemisphere and AAV is delivered to non-tumoral astrocytes.

We agree that there is likely some effect within the brain at large to AAV transfection given the ability of chemokines to stimulate recruitment to a larger general area than their site of production. We have assessed this utilizing our intracranial delivery model wherein we administered AAV6-CXCL9 to either the direct tumor site or to contralateral hemispheric tissue. We did see a trend towards observed treatment effect with opposite hemispheric injection, potentially opening the door for this therapies’ application to more difficult-to-treat locations or those with diffuse disease. However, when we applied statistical analysis to these treatment groups there was no significant improvement in survival in the contralateral transfection group compared with sham control injection. We have included this interesting finding in our Extended Data Figure 5f. Relevant to this query, we did find that in tumors given ‘delayed treatment,’ whole brain imaging of experimental tumors revealed significant tumor dissemination into the contralateral hemisphere by way of ventricular spreading (Extended Data Figure 5h). In this regard, contralateral delivery of AAV6 could conceivably function in a prophylactic

capacity, a concept experimentally tested by another research group (Maguire, C.A., *et al.* Genetic modification of neurons to express bevacizumab for local anti-angiogenesis treatment of glioblastoma. *Cancer Gene Ther* **22**, 1-8 (2015)).

The manuscript amendment to address this concern can be found on **Page 5, Lines 253-258**, as well as in **Extended Data Figure 5f**.

(2) Figure 3c. There was no competitive co-culture comparing the migratory capacity of T cells towards CXCL9-expressing GL261 and astrocytes depicted.

We thank the reviewer for this interesting suggestion. To address this, we have performed an expanded competitive co-culture chemotaxis assays to identify whether or not there is an advantage for either tumor cells or astrocytes following AAV6-CXCL9 transduction in promoting the migratory capacity of T cells. What we ultimately found was that there is no strategic advantage in terms of directing preferential chemotaxis of T cells, and that T cells migrated towards each AAV6-CXCL9 transduced tumor and astrocyte cells uniformly (Extended Data Figure 5a). Interestingly, this occurs despite a few inherent physiological differences between these two cell populations. Namely, when assaying for CXCL9 production by each astrocytes and tumor cells immediately prior to adding the lymphocytes to the assay, we found that the chamber housing the tumor cells had ~2x the detectable level of CXCL9 (Extended Data Figure 5b). We surmised that this could likely be due to the inherent differences in proliferative capacity, and indeed found that the cell doubling time of GL261 tumor cells was greater than that of the C8-D1A astrocyte line (Extended Data Figure 5c). In this sense, it's likely that tumor cells are capable of producing more transgene expression in an acute phase until the transgene is diluted amongst daughter cells. What we concluded, is that there could be a maximum biological threshold of CXCL9 concentration to facilitate lymphocyte chemotaxis, beyond which there is no added advantage. Moreover, this also reaffirms that lymphocyte responsiveness to CXCL9 chemokine production occurs autonomously of the cellular source, where any transduced cell, be it a tumor cell, astrocyte, or other cell can serve as a homing beacon for T cell recruitment. However, as we point out in the manuscript, the *in vivo* tropism of AAV6 in preclinical GBM models is the astrocytes surrounding the tumor and injection site. We feel this is an advantage to the treatment paradigm given the comparatively reduced proliferative capacity of these cells and potential for more durable transgene expression, yet welcome any transfection of tumor as added benefit.

The manuscript amendment to address this concern can be found on **Page 4, Lines 184-191**, as well as in **Extended Data Figure 5a, b, and c**.

(3) Figure 3e-h. Anti-PD1 should unleash peripheral and intratumoral T cells and thus T cell proliferation. The authors measure T cell abundance in tumors but not T cell infiltration. This should be carefully taken into consideration.

We agree with the reviewer's comments on the number of tumor infiltrating T cells in terms of abundance does not always correlate with the depth, degree, or location of infiltration. To better pair these interesting flow cytometry findings we have performed 3D confocal microscopy of control and combination treated tumors analyzing T cell infiltration. As shown in the newly added Figure 3i, we observed both enhanced T cell accumulation as well as dissemination deeper into tumors following combination therapy of AAV6-CXCL9 and anti-PD-1 therapy.

The manuscript amendment to address this concern can be found on **Page 4, Lines 205-210**, as well as in **Figure 3i**.

(4) Treating mice at day 5 is very early, at this time point, in e.g., preclinical MRI, tumors are macroscopically not visible. How is intratumoral injection confirmed then? . The authors should repeat their therapeutic treatments using AAV6-CXCL9 at day 10-12.

The reviewer brings up a good point and challenge with the preclinical modelling of GBM in murine hosts. When tumors are allowed to establish for too long it raises difficulty in treatment time to efficacy before safe humane endpoint from disease burden in aggressive models, and when treated too early it risks over stating the treatment effect. 5 days was chosen based on well established published models on tumor injection to treatment timelines. To overcome the limited size of tumor

at this stage, the administration of vaccine through the surgical implantation field has alleviated many of the concerns of tumor targeting. As these coordinates are shared between needle implantation of tumor cells and inoculation of vaccine, the tumor falls within the treatment field in the majority of animals. This is demonstrated through the tumor-tropic nature of AAV6 fluorescent reporter detection originally shown in Figure 2a and b following the day 5 AAV intratumor injection strategy.

There is always potential for aberrant positioning of the animal, spread or dissemination of tumor outside the inoculation site, and needle angle playing a part in distribution of vaccine. In the murine tumors analyzed under confocal microscopy we regularly see evidence of needle path effect with linear parallel cuts through the tumor. We additionally have performed these experiments utilizing these shared coordinates from multiple research personnel with reproduction of the on treatment effects and results of control vs treated animals. We agree this is by no means a perfect representation of identical treatment between animal subjects, but based on statistical analysis even with these differences between subjects, the results remain significant where indicated.

As stated we agree that there is no perfect time point for initiation of therapy in murine GBM models, and to some 5 days is too early in tumor development. There are other reported models of therapeutic intervention where treatments are initiated even earlier in tumor development, and also in the prophylactic setting. To examine treatment efficacy at later stage tumors, we have replicated these treatments in mice, allowing for 12 days of tumor growth prior to intervention with vaccination plus anti-PD-1 ICB. In these mice, there is again a trend for improved survival even when delaying treatment, but this was significantly reduced compared to animals starting treatment at day 5 (Extended Data Figure 5g). Accompanying these experiments, we performed whole brain imaging studies to examine the pattern of AAV6 fluorescent reporter transduction when delivered in a delayed fashion. We found that in these larger tumors, AAV6 expression was confined to tumor within the injection path (which can be visualized at this large scale, corroborating our ability to perform intratumor injections based on stereotactic coordinates). Although the AAV6 transduction field was sizeable for a single injection, we found evidence of broad disease dissemination in several animals, extending into the contralateral hemisphere well beyond the treatment site. An example of this is shown in Extended Data Figure 5h. Treatment of large bi-hemispheric tumors would be a challenge in any clinical setting, and a more realistic approach for clinical translation may involve post-resection treatment, intratumoral delivery into localized recurrences, and/or the use of delivery systems that allow for larger geometric coverage of invasive gliomas.

Based on this finding, and similar to the reviewer's recommendations on vaccine administration being somewhat blind, we think that a modified vaccination strategy will likely be necessary for larger tumors that involves multifocal delivery. At current, delivery is administered centrally to the tumor at multiple depths treating deep to superficially along the needle track. Future plans include developing enhanced surgical methods of performing circumferential and central vaccine inoculation for the treatment of larger experimental tumors including a larger surgical window through burr hole or partial craniectomy (protected by a bone or glass flap) to distribute the vaccine at multiple sites in the tumor. We are also optimizing a tumor resection model to experimentally mimic disease recurrence. We are excited by the potential for a multi-site administration to enhance the vaccines' efficacy that also fits with current clinical delivery of oncolytic viral therapy for solid tumors.

The manuscript amendment to address this concern can be found on **Page 5, Lines 258-266**, as well as in **Extended Data Figure 5g and 5h**.

(5) Fig 4c/d: Where are the biological controls (AAV6-GFP?) for the IF?

For this experiment, we utilized a sham control given the degree of spectral overlap observed between EGFP (used as the AAV transgene control) and EYFP produced by the murine transgenic reporter model. The figure panels representing this control have been added to **Figure 5c**.

(6) What is the role of CD4+ T cells when applying AAV-CXCL9?

This is a great question and something we had thought of while performing our CD8 lymphocyte depletion assay. To address this, we performed a CD4 depletion experiment in similar fashion to the CD8 depletion experiment. In this survival study we observed that CD4 depletion reduced the overall efficacy of combination AAV6-CXCL9 plus anti-PD-1 ICB treatment, yet extended survival was still observed in a subset of the mice. Interestingly, CD4 depletion also accelerated time to humane endpoint in the control arm, implying the need for coordinated immune response to combat tumor progression.

The manuscript amendment to address this concern can be found on **Page 5, Lines 239-245**, as well as in **Figure 4g and Extended Data Figure 5e**.

(7) It would be important to assess, how CXCL9-expressing astrocytes interact with T cells.

We agree with the reviewer's comments and have utilized high resolution confocal microscopy to evaluate these interactions between T cells and astrocytes. In mice receiving AAV6-CXCL9 plus anti-PD-1 ICB combination treatment, we identified regions of T cell accumulation within astrocyte networks within and around tumor. Voxel-based co-localization between GFAP (astrocytes) and EGFP (T cells) was used to identify areas of direct cell-cell contact. We found this phenomenon to be quite common as shown in the newly added Figure 3j, and indicates the possibility of a direct cellular communication between astrocytes and T cells. It is an area of great interest and future investigation. The prospect of astrocytes functioning in a pro-inflammatory manner, for example by serving as antigen presenting cells, or conversely by impairing the function of T cells through means of deactivation or sequestration has not been examined in the context of brain tumors, and could produce an entirely new avenue for developing targeted therapeutics.

The manuscript amendment to address this concern can be found on **Page 4, Lines 210-215**, as well as in **Figure 3j**.

Reviewer #3 (Remarks to the Author): with expertise in AAV-mediated gene therapy of the brain

CXCL9 recombinant adeno-associated virus (AAV) virotherapy sensitizes glioblastoma (GBM) to anti-PD-1 immune checkpoint blockade by Roemeling et al. seeks to develop a novel AAV-based therapy using tumor stromal cell expression of the chemokine CXCL9 in combination with anti-PD-1 antibodies for glioblastoma (GBM) therapy. The authors tested several capsids for transduction of glioma cells in culture and AAV6 was one of the best performers. Intratumoral injection of AAV6-GFP into mice bearing GBM tumors, yielded some transduction of tumor cells, but the vast majority of GFP positive cells for astrocytes around/within the tumor. The authors then tested the use of AAV-CXCL9 as a therapeutic agent for GBM. In vitro assays demonstrated chemotaxis of T cells towards AAV-CXCL9 transduced tumor or astrocytes. In vivo, low levels of CXCL9 could be detected in the tumor/AAV-CXCL9 injected side of the brain, but not the contralateral side (suggesting a relatively low level of transduction). In two glioma models, the authors also found that AAV6-CXCL9 combined with anti-PD-1 antibody increase CD8 T cell recruitment to the tumor. Next the authors performed a therapeutic study with AAV6-CXCL9 combined with anti-PD-1. A modest but significant increase in median survival was observed in two glioma models with the combined treatment over controls, with long-term survivors (50% and 25%) observed in both models. They also found that CD8 T cells were crucial to the survival using a CD8 depletion experiment. The authors next performed extensive RNAseq analysis to observe differences in specific cell type signatures between the different control groups and made some interesting observations. The in vitro and in vivo AAV cancer gene therapy experiments have been done very well, and the reviewer is appreciative of the amount of labor that went into this work. Also, the strategy of using AAV vectored-chemotaxis to recruit T cells into the tumor environment along with anti-PD-1, seems like a feasible strategy, and the authors provide proof of concept data of efficacy.

This notwithstanding, a major novelty of the paper appears to be the observation that "AAV targeting of tumor-associated cells may carry several distinct advantages over directly targeting cancer cells...." This statement appears to suggest that the authors were discovering/proposing something novel, while it is not (see comment 1 below). Unfortunately, the existing literature (which includes AAV transduction of astrocytes for GBM therapy) greatly reduces the novelty of the work for this journal's broad audience. It does appear, however, that the combined use of AAV-based chemokine expression to bring T cells into the tumor adds some novelty to this study.

Comments:

Comment 1: This strategy of using the tumor stromal microenvironment for expression of AAV-vectored transgenes has been used by other groups (see below) successfully in past published work, however the authors did not appear to be aware of this. Proper citations of the prior work should be carefully assessed by the authors.

Examples:

We thank the reviewer for their knowledge of the AAV vaccine design landscape for CNS malignancies. We also acknowledge that the field of study for the therapeutic utilization of AAV to treat a variety of CNS diseases, including malignancy, is vast. Numerous serotypes, gene cargoes, delivery methods, etc. have been explored, and citing each and every one of these is a challenge. Having reviewed the citations provided below, we have incorporated these into the body of the manuscript where appropriate. We also note in the manuscript the key differences between our work and that of the published literature.

<https://pubmed.ncbi.nlm.nih.gov/36217021/>

In this manuscript, Yao *et al.* describe their work generating blood brain penetrant variants in capsids AAV8 and AAV9 (whereas we utilize capsid AAV6), allowing for systemic delivery of encoded transgenes resulting in enhanced brain expression in mice and non-human primates. While this is an excellent strategy for producing transgene signal in the brain without the use of invasive delivery methods (i.e. intratumor, intraparenchymal), we believe that direct tumor inoculation may be required for delivering chemotaxis-based gene cargo as is the case with our therapeutic strategy. While Yao *et al.* encouragingly show that their engineered AAV.CPP.16 capsid is capable of transducing preclinical tumor cells *in vivo*, the encoded reporter transgene is also visibly and broadly expressed in non-diseased normal tissue, and includes both neurons and astrocytes. Given that CXCL9 is a call-and-receive signal for T lymphocytes, off-target systemic transduction (i.e. liver, skeletal muscle, heart, and dorsal root ganglion as observed by Yao *et al.*) could dilute the ability of T cells to traffic to the brain tumor. Moreover, off-target expression of CXCL9 in non-diseased brain tissue could inadvertently provoke unwanted inflammation in these areas, warranting concerns towards safety. In considering non-chemokine based gene cargoes, the prospect of using brain-tropic AAV serotypes holds immense promise, as is demonstrated by Yao *et al.* where they are able to show remarkable survival responses following the delivery of a PD-L1 immune checkpoint inhibitor to GL261 tumors. Broad transgene signal in the brain should also be considered for diffuse CNS tumors, such as primary CNS lymphoma, metastatic CNS tumors (e.g. melanoma, breast), or diffuse spread of primary brain tumors.

This reference has been added to both the introduction, as well as the discussion where we dialogue the pros and cons of leveraging brain-tropic AAV serotypes for the delivery of lymphocyte specific chemokines on **Page 10, Lines 500-507**.

<https://pubmed.ncbi.nlm.nih.gov/37172581/>

Similar to the previous reference by Yao *et al.*, here Ramachandram *et al.* examine the therapeutic efficacy of a novel brain-targeting AAV serotype that is delivered systemically via intravenous injection, based on a modified AAV2 mutant capsid produced by Dr. Martin Trepel's research group that preferentially transduces brain endothelial cells (reference: <https://www.embopress.org/doi/full/10.15252/emmm.201506078>). In reviewing the latter reference, this modified capsid, while showing a high predilection for targeting CNS tissues, also produces off-target transduction in heart, lung, liver, kidney, and muscle tissue following intravenous injection. While potentially useful in delivering a different transgene cargo, peripheral expression of CXCL9 could deter selective CNS homing of CD8 T cells, diluting the therapeutic response to treatment. Whole brain transduction of endothelial cells could also prompt unwanted neuro-inflammation in normal healthy CNS tissue. In the research article by Ramachandram *et al.* they examine the impact of AAV encoding LIGHT (TNFSF14) on enhancing/inducing the formation of high endothelial venules and tertiary lymphoid structures to promote T cell recruitment to the CNS. Interestingly, the authors found that TLS were predominantly found in close proximity to meningeal tissues surrounding the cortex or the ventricles, however it wasn't clear if there was increased intratumor (parenchymal) penetration by T cells as a result of AAV-LIGHT treatment. It would be interesting to evaluate the efficacy of AAV-LIGHT

against tumors that spread into the leptomeningeal space, including metastatic solid tumors (i.e. melanoma), or primary brain tumors that demonstrate preferential invasion into this region of the CNS (i.e. medulloblastoma).

This reference has been added to both the introduction, as well as the discussion where we dialogue the pros and cons of leveraging brain-tropic AAV serotypes for the delivery of lymphocyte specific chemokines on **Page 10, Lines 500-507**.

<https://www.ncbi.nlm.nih.gov/pmc/articles/PMC2863297/>

In this reference, Maguire *et al.* produce a strong argument for targeting normal residing tissues in the brain instead of tumor targeting, and stress the importance of AAV serotype selection in establishing the source of transgene production. In their research article, they leverage AAVrh.8 to deliver hIFN- β in the prophylactic setting to prevent the growth of stereotactic implanted tumors. What's particularly interesting about this study, is that it is performed using immune compromised athymic nude mice and human tumor cells, indicating that the protective anti-tumor activity of IFN- β is likely facilitated by the innate immune system, although this isn't discussed in great detail in the manuscript.

We have included this reference and small discussion in the revised manuscript on **Page 9, Lines 448-452**.

<https://www.ncbi.nlm.nih.gov/pmc/articles/PMC6454875/>

In this article, Volak *et al.* demonstrate good modification potential of systemically delivered exo-AAV9 therapy with enhanced specificity for myeloid cells or astrocytes. Their transgene cargo, IFN- β , was able to produce a mild survival response. While the authors were able to show that their modified exo-AAV9 construct had significantly reduced 'off-target' transduction in liver, it is possible that there could be off-target transduction in other tissues, which may have diluted the anti-efficacy of this treatment strategy. We agree with their findings that targeting of astrocytes or microglia within the tumor has high therapeutic potential, particularly towards modifying the immunological landscape of GBM, and have added this reference to our modified manuscript.

<https://pubmed.ncbi.nlm.nih.gov/25501993/>

There are some significant and interesting differences between our research and the findings published by Hicks *et al.* We agree with the authors' suggestion of intracavitary AAV delivery post-resection as a strategic window of opportunity to directly apply the vaccine to the tumor site. Although the authors found that their AAVrh.10 construct selectively targeted neurons following intratumor and intraparenchymal injection (where in our study AAV6 preferentially transduces astrocytes), it aligns with our treatment paradigm of direct tumor delivery and selective transduction of non-tumor cells within the tumor microenvironment. Here, the authors were able to demonstrate that AAV induced expression of a secreted bevacizumab transgene to modulate tumor angiogenesis conferred a survival advantage in immune-compromised humanized models of GBM, whereas our study focuses on the ability of secreted CXCL9 to promote immune-mediated anti-tumor activation. This reference has been added to our modified manuscript.

Comment 2: It seems that the AAV6 transduction of tumor-associated astrocytes was an unexpected observation by the authors and that expression levels of CXCL9 by this system were quite low. To improve the outcome of the combination therapy, it is likely that the system needs further optimization at the capsid and transgene expression cassette levels.

The reviewer is correct in pointing out that our identification of astrocytes as the primary driver of AAV6-CXCL9 transfection and CXCL9 production *in vivo* was not initially anticipated as part of the therapy's pre-designed nature. During our extensive AAV serotype screening, we observed excellent tumor cell transduction with several capsids across various primary human glioma cell models. While we had assumed that, in addition to glioma cells, we might observe transduction of other tumor microenvironment constituents (such as microglia, astrocytes, and neurons) following *in vivo* delivery, we were surprised by the significant uptake and transgene production by astrocytes, with minimal expression in tumor cells.

Upon making these observations, we considered numerous potential paths forward, including retooling our AAV constructs with different promoters to achieve cell-specific expression. However, instead of pursuing that route, we chose to capitalize on our initial findings, which demonstrated robust transgene expression in a subset of cells located proximally within the tumor microenvironment. We deliberately selected CXCL9 as our transgene cargo due to its properties as a secreted

therapeutic transgene capable of conferring anti-tumor activity in a cell-autonomous manner. Furthermore, CXCL9 is known for its role as a lymphotactic chemokine, and since lymphocyte recruitment into GBM tumors is a well-established barrier to immunotherapy response, its inclusion held additional strategic value.

We also believe that the expression level of CXCL9 within our system reached biologically meaningful levels, and its direct contribution to this production results in significant survival benefits for combination-treated animals. However, we acknowledge that additional modification of the viral capsid and transgene cassette may enhance the therapeutic efficacy of our treatment strategy, as is typical in any proof-of-concept study.

Finally, extensive capsid engineering needs to be balanced with the scope of murine studies versus clinical translation and ultimately human trials. Previous strategies of AAV viral therapy have involved significant alteration and modification in murine hosts, only to encounter issues such as human subjects clearing these modified viruses through peripheral immune activation, or the same modifications yielding different biological transduction in human tissues. To address these concerns, we plan to continue our modification steps using human-derived brain slice modeling and in silico modeling predictions. These approaches aim to alleviate some of the challenges associated with cross-species modifications.

Reviewer #4 (Remarks to the Author): with expertise in glioblastoma, cancer immunology

This is an interesting and timely study from a well-established brain tumor group at the University of Florida. It is well-known that malignant gliomas have greatly reduced numbers of tumor infiltrating lymphocytes (TIL), but yet these cells are necessary for functional anti-tumor immunity and immunotherapy. Van Roemeling, et.al, try to address this significant gap through the AAV recombinant production of CXCL9, a potent chemokine known to recruit T cells into the tumor microenvironment (TME). While CXCL9 by itself, does not seem to be able to elicit significant migration of T cells into the TME, in conjunction with PD-1 mAb blockade, these investigators do see elevated CD8+ T cells in the tumor, and was associated with extended survival in mice bearing two different models of malignant gliomas. The efficacy of the combination treatment was completely dependent on CD8 cells. Using several high dimensional technologies (e.g. single cell RNAseq), this group also showed how the combination of AAV-encoded CXCL9 + PD-1 mAb resulted in enhanced T cell activation, upregulation of immune cell exhaustion and cellular crosstalk with other tumor infiltration immune cells. Such pre-clinical studies form the basis of a better understanding of how TIL density is limiting in this kind of tumor, and outline a clinically translatable strategy to target this gap. While significant and important for the field, a few issues ought to be clarified to best articulate the overall message and interpretation of the data. These issues are outlined below:

Major Comments.

1. Fig 2. It is interesting that this AAV preferentially infects astrocytes in vivo. Is that really the cell type that they would want to move forward with and use? I would think that cells of the myelo-monocytic lineage would be more effective to target, as they are already known to secrete these kinds of chemokines during productive anti-tumor immune responses. Did the authors ever identify different AAV's that could infect myeloid cells?

We agree with the reviewer that the finding of astrocytes as the predominant cell affected by transfection was intriguing. We chose AAV6 in part due to its ability to transfect immune cells within the brain parenchyma, particularly microglial cells, as previously reported by other research groups (Rosario *et al.* Microglia-specific targeting by novel capsid-modified AAV6 vectors. *Mol Ther Methods Clin Dev.* 2016; 3: 16026; Maes *et al.* Optimizing AAV2/6 microglial targeting identified enhanced efficiency in the photoreceptor degenerative environment. *Mol Ther Methods Clin Dev.* 2021; 23: 210-224; O'Carroll *et al.* AAV targeting of glial cell types in the central and peripheral nervous system and relevance to human gene therapy. *Front Mol Neurosci.* 2021; 13 - 2020). However, in our experiments, we did not observe transfection of either microglia or tumor-associated myeloid cells, as demonstrated in Extended Data Figure 3.

While we acknowledge the significant therapeutic potential of direct microglial transduction in reshaping the glioma immune microenvironment, considering that CXCL9 is a secreted immunologic protein, there may be less of a requirement for a true antigen-presenting cell (APC) to be the recipient of transduction. Additionally, astrocytes have been reported to be co-opted by tumors to produce and secrete other soluble proteins such as interleukins, serving as mechanisms of

proliferation and environmental remodeling. Therefore, astrocytes possess the necessary machinery and capacity for chemokine production, making them suitable for this approach.

Furthermore, other research groups have focused on glial fibrillary acidic protein (GFAP) specificity of AAV cassette inserts to facilitate transfection, aiming to create a tumor barrier at the "glial scar" or area of reactive glial cells surrounding the tumor (Maguire et al.). Considering these points, we believe that targeting astrocytes for CXCL9 production holds potential in this disease space.

2. Fig 3. It would be useful to understand better how the in vitro migration assay was set up? This is not articulated well in the text or the legend.

Understanding our experimental approach is integral for disseminating our research findings and enabling others to either reproduce our data or utilize this methodology to enhance their own research objectives. While it may be difficult to ascertain the exact approach from the results and the figure legend, we have included a very detailed methodological approach for this assay in the Materials and Methods section of the manuscript. This can be found on **Page 37, Lines 1000-1014**.

3. Fig. 3. The authors nicely show an increased CD8+ TIL infiltration into two distinct pre-clinical models of glioma with AAV-CXCL9 + PD-1 mAb. It would be important, though, to show the receptor expression on these CD8 T cells to argue that the CXCL9 was acting through its cognate receptor. Do particular T cell subsets express CXCR3?

This is an excellent suggestion posed by the reviewer. To parse this out, we leveraged our scRNAseq data to initially determine which CD45+ immune clusters express *Cxcr3* transcripts. We found that lymphocyte subsets (CD8+ T cells, CD4+ T cells, NKT cells) predominantly express *Cxcr3*, with limited and sporadic *Cxcr3* expression detected across the other immune clusters. We also found that treatment did not change the affinity of *Cxcr3* transcript expression in other immune cell clusters, as it was still focused in lymphocytes. This newly added data can be found in **Figure 5h** and **Extended Data Figure 7e**. Next, we subdivided lymphocytes into 11 lymphocyte specific clusters as shown in **Figure 5i** and **Extended Data Figure 7e**. Of these, we found that *Cxcr3* was uniformly expressed across nearly all clusters, with the exception of 2 populations (CD16+ CXCR3low CD8 T and CXCR3low Tregs), which had slightly reduced *Cxcr3* transcript expression as shown in the newly added **Figure 5j**. We have also modified the text on **Page 6, Lines 285-293** to accommodate these new findings. These data corroborate that lymphocytes are the likely target of CXCL9 AAV gene therapy in terms of its known chemotactic activity.

4. Fig 6. It is difficult to understand what question and/or new information the Nanostring adds to this story? It is also difficult to visually understand the significant changes in Fig. 6f. Maybe better just to highlight or focus on the selected cell type gene expression changes? In addition, it is not clear how these studies were done? I don't see any methods for the Nanostring analyses and whether they used single cell suspensions or bulk tissue?

In this experiment, we leveraged the NanoString analyses to better interpret transcriptional changes in cell populations from our single cell RNA sequencing data in response to single and combination AAV6-CXCL9 gene therapy with anti-PD-1 ICB, as it provides clearer insight towards specific pathway activation and subsequent phenotype analysis as opposed to quantifying individual transcripts. We also wanted to include the global changes across all CD45+ cells isolated across treatment groups, as shown in Figure 6f to provide an expansive summary of each immune cell cluster. To the reviewer's point, it is difficult to deconvolute meaningful differences from the heat map, and so this is why we included the additional panels in **Figure 6g-l**. These represent significant changes in the pathways included in the NanoString analysis specifically in the CD8 T cell population which is required for anti-tumor efficacy of the combination therapy. We have modified the text in the Results to try and make the connectivity between the heatmap and the individual graphs clearer on **Page 7, Lines 339-342**.

The detailed methodological approach for the scRNAseq and Nanostring analysis can be found beginning on **Page 37, Lines 1016-1059** in the Materials and Methods section of the manuscript. We have edited the sub-header, which previously read as "Single Cell RNA Sequencing, Quality Control, and Data Analysis" to "Single Cell RNA Sequencing, Quality Control,

NanoString and Data Analysis” to make it more evident that this information can be found in this sub-section of the manuscript.

5. Fig. 7. Same basic methodology question here. How were these studies done? It is not clear from the text.

This information can be found in the Materials and Methods section on **Page 36, Lines 928-941**. On request, we could include greater detail regarding methodology outside of the methodology section to improve ease of reader interpretation if this is felt to still be unclear. This has not been expanded in this current submission due to sheer volume of the manuscript.

REVIEWERS' COMMENTS

Reviewer #1 (Remarks to the Author):

In the revised manuscript, all of my concerns in the previous review have been well addressed. The errors in the figures and figure legends have been fixed.

Because this study demonstrates the positive therapeutic effect of the proposed strategy and discovers the immunological signatures related to the effect, I believe that the study will bring new idea and application to the field of the combination therapy.

Reviewer #2 (Remarks to the Author):

The authors provide a substantially revised version of their manuscript and have conducted additional in vivo experiments further corroborating their findings where requested. I do not have any further comments or concerns.

Reviewer #4 (Remarks to the Author):

This revised submission is better and has answered the main questions that were posed in the original draft. In particular, the finding that astrocytes are the main parenchymal population transduced with the AAV, and are actually capable of secreting biologically relevant concentrations of CXCL9 is innovative and unexpected. Furthermore, they also included single cell data that shows how CD4+ and CD8+ T cells, along with NKT cells, specifically express the particular chemokine receptor (CXCR3) that responds to the chemokine. I do not have any further questions.